# On-Demand Drug Delivery Systems Using Nanofibers

**DOI:** 10.3390/nano11123411

**Published:** 2021-12-16

**Authors:** Baljinder Singh, Kibeom Kim, Myoung-Hwan Park

**Affiliations:** 1Department of Convergence Science, Sahmyook University, Seoul 01795, Korea; vishalmasih94@gmail.com; 2Convergence Research Center, Nanobiomaterials Institute, Sahmyook University, Seoul 01795, Korea; kibumsy@syu.ac.kr; 3Department of Chemistry and Life Science, Sahmyook University, Seoul 01795, Korea; 4N to B Co., Ltd., Business Incubator Center, Hwarang-ro, Nowon-gu, Seoul 01795, Korea

**Keywords:** on-demand drug release, nanofibers, electrospinning, polymers, drug loading, drug administration

## Abstract

On-demand drug-delivery systems using nanofibers are extensively applicable for customized drug release based on target location and timing to achieve the desired therapeutic effects. A nanofiber formulation is typically created for a certain medication and changing the drug may have a significant impact on the release kinetics from the same delivery system. Nanofibers have several distinguishing features and properties, including the ease with which they may be manufactured, the variety of materials appropriate for processing into fibers, a large surface area, and a complex pore structure. Nanofibers with effective drug-loading capabilities, controllable release, and high stability have gained the interest of researchers owing to their potential applications in on-demand drug delivery systems. Based on their composition and drug-release characteristics, we review the numerous types of nanofibers from the most recent accessible studies. Nanofibers are classified based on their mechanism of drug release, as well as their structure and content. To achieve controlled drug release, a suitable polymer, large surface-to-volume ratio, and high porosity of the nanofiber mesh are necessary. The properties of nanofibers for modified drug release are categorized here as protracted, stimulus-activated, and biphasic. Swellable or degradable polymers are commonly utilized to alter drug release. In addition to the polymer used, the process and ambient conditions can have considerable impacts on the release characteristics of the nanofibers. The formulation of nanofibers is highly complicated and depends on many variables; nevertheless, numerous options are available to accomplish the desired nanofiber drug-release characteristics.

## 1. Introduction

The development of on-demand drug delivery systems (DDSs) has received significant attention owing to the high demand for controlled delivery of various drugs to organs and cell tissues [1,2]. The aim of advanced and smart DDSs is to achieve high therapeutic effects while avoiding adverse side effects on human health [3,4]. Most conventional DDSs are administered via enteral routes, such as tablets, capsules, and granules, while others are administered via parenteral routes, such as intravenous, intra-arterial, intramuscular, or subcutaneous routes [5,6]. There are various drawbacks to the routes and methods of administration, such as first-pass metabolism and discomfort [7,8].

The drugs that reach the market have potential side effects; for example, anticancer chemotherapeutics are a source of concern for both therapists and patients because of their inherent toxicity [9,10]. Although their efficacy and target selectivity have increased over time, significant side effects, such as infections, vomiting, tiredness, loss of taste, anemia, and immune system damage, still exist. The mode of administration affects the therapeutic benefit of a medication by influencing various aspects, such as pharmacokinetics, distribution, pharmacodynamics, metabolism, and toxicity [11,12]. With the discovery of nanomaterials, novel methods have been developed for preparing DDSs [13,14]. Using passive or active targeting techniques depending on the final formulation, nanomaterials can be utilized as carriers to wrap and distribute drugs that are extremely toxic, insoluble, rapidly cleared, or unstable as free molecules [15,16,17,18,19,20,21,22].

Among several alternatives, nanofibers (NFs) have attracted much attention because of their immense potential for smart DDSs (Figure 1) [23,24]. NFs offer unique qualities, such as a microstructure that is comparable to that of an extracellular matrix; a large surface area; high porosity with interconnectivity, which enhances cell adhesion, proliferation, drug delivery, and mass transport capabilities; and a variety of matrix materials [25,26]. Furthermore, the advancement of specialized electrospinning methods has provided new possibilities for the loading and release of insoluble drugs [27,28]. Excellent stability, improved targeting, low toxicity, high drug-loading capacity, remarkable mechanical characteristics, encapsulation of a wide range of medicines, and compatibility with thermolabile medications are all benefits of NF scaffold formulations. Delivering drugs to patients in the most appropriate manner has long been a challenge [29,30]. As delivery matrices, a wide range of biocompatible, biodegradable, and nonbiodegradable polymeric materials can be employed [31].

NFs for therapeutic applications follow several basic designs. Homogeneous architectures, in which the drug is spread throughout the polymer matrix, and core-shell NFs, in which the drug-carrying matrix is purified by polymers, are the most common designs. NFs stabilize with active molecules on their surface [32,33,34]. Because of their benefits, such as improving drug solubility and bioavailability or regulating the pace and location of administration, rapid dissolution and controlled release have become crucial for creating innovative drug delivery methods [35]. NFs have several properties, such as protection of drugs from systematic decomposition in the blood circulation, controlled release of the drug at a constant rate over a longer period of time, drug release only at the targeted body area, and permeation of certain membranes or barriers [36].

Several review studies have been published to highlight the significance of NFs in DDSs for the treatment of many complex diseases [27,29,37]. Following recent developments, we review the most recent findings and methods for achieving on-demand release of drugs, convenient mechanisms of drug loading, and various modes of drug administration. In this study, we report the preparation, effects, and applications of electrospun NFs as on-demand drug delivery platforms [38]. Many pharmaceutical polymer excipients are often employed in the development of novel DDSs [39]. The use of electrospinning in conjunction with pharmaceutical polymers provides innovative techniques for creating novel DDSs and modifying the electrospinning process may allow flexibility for changing the characteristics of DDSs [40]. This review also discusses several other related challenges and potential remedies.

## 2. Types and Preparation of NFs as DDSs

NFs are typically produced as nonwoven mats using the electrospinning process. Electrospinning is a manufacturing technology that uses an electrostatically driven process to generate electrospun fibers as shown in Figure 2. The diameter of these fibers generally ranges from tens of nanometers to a few micrometers. One of the primary advantages of the electrospinning technology is its processing flexibility, allowing for the creation of fibers with a variety of arrangements and morphological features. The popularity of the electrospinning process has allowed many technologies to develop and evolve over the last decade, including tissue engineering, regenerative medicine, and encapsulation of bioactive compounds. DDSs that employ the electrospinning approach have higher control and predictability than conventional methods.

A single spinneret is used in typical electrospinning to fabricate NF mats because it is a convenient and cost-effective process [40]. However, it lacks high throughput, which is a critical requirement for practical and industrial applications. The multi-tip system also necessitates a large amount of space for the equipment, which raises the cost of the manufacturing process [41]. To address this issue, needleless electrospinning has been identified as an effective method for increasing the continuity of the fabrication process and productivity, as an electrical force is applied directly to the liquid surface without the use of a needle nozzle [42].

A wide range of polymers have been used to fabricate NFs, including poly (N-isopropylacrylamide) (PNIPAM), polylactic acid (PLA), poly-L-lactic acid (PLLA), PLLA/hydroxyapatite (PLLA/HAp), polyglycolic acid (PGA), polycaprolactone (PCL), polycarbonate, polyurethane (PU), polyethylene glycol (PEG), poly(D,L-lactide-co-glycolide) (PLGA), PLGA/gelatin, poly(ε-caprolactone) (PCL), polyethylene oxide, poly(L-lactide-co-caprolactone), polyvinyl alcohol (PVA), and polyvinylpyrrolidone (PVP). However, not all of them can be employed as medication delivery methods [39,43]. Among several polymers, PLLA, PLA, PLGA, PGA, PCL, and PEG are used to fabricate NFs for drug release [39,44]. These polymers are well known because they are sufficiently strong to be electrospun, biocompatible, and beneficial for cell adhesion [45].

### 2.1. Blended NFs

Single-phase material structures are among the most frequently produced blended NFs [46]. Single-phase fibers consist of only one structure, which can be composed of, but not limited to, one material or a combination of one material and a bioactive component [47]. An appropriate technique for drug loading should be devised based on the characteristics of the drug to achieve optimal drug release kinetics [48]. Prior to electrospinning, drugs can be simply dissolved or distributed in a polymer solution, or in rare circumstances, physically or chemically attached to an NF surface [49]. According to Nakielski et al. [50], electrospun NFs with the capacity to respond to near-infrared (NIR) light stimulation are vital for creating highly efficient biomedical nanoplatforms with a polytherapeutic approach. Their polymer solution for electrospinning was prepared by dissolving PLLA in a 9:1 (*w*/*w*) mixture of CHCl_3_ and N,N-dimethylformamide (DMF). Afterward, rhodamine B (RhB) was combined with the polymer solution (1 wt% with respect to the polymer). RhB was selected as a model drug because it is soluble in both polymer and aqueous solutions. The PLLA NFs were electrospun at a positive voltage of 17 kV and a flow rate of 800 µL/h. The fibers were collected using a grounded rotating drum collector. During electrospinning, the temperature was approximately 23 °C, and the humidity ranged from 50 to 60%. This NF film containing RhB as a model drug was further modified with a hydrogel precursor solution containing N-isopropylacrylamide (NIPAAm), NIPMAAm, and BIS-AAm in a proportion of 1:0.05:0.03, respectively, with dispersed gold nanorods (GNRs) to obtain the mutual effects of temperature rise, water movement that triggers the release of molecules from the NFs, and photothermal therapy.

Kim et al. [51] successfully fabricated a smart hyperthermia NF using a blended electrospinning technique to achieve switchable drug release for inducing cancer apoptosis (Figure 3). In their study, a poly(NIPAAmco-HMAAm) solution was prepared by dissolving the copolymer in 3 mL of 1,1,1,3,3,3 hexafluoro-2-propanol (HFIP). Magnetic nanoparticles (MNPs) and doxorubicin (DOX) were dissolved in distilled water. The copolymer and MNP/DOX solutions were combined in a 10:1 ratio to obtain the final weight fractions of MNPs and DOX in the polymer. Their study demonstrated switchable variations in the swelling ratio in response to alternating on–off switches of an alternating magnetic field (AMF), because the self-generated heat from the integrated MNPs causes the deswelling of polymer networks in the NF. In response to the AMF, an on–off release of DOX from the NFs was observed. Jariwala et al. [52] reported the concept of utilizing electrospun piezoelectric NFs as an on-demand drug delivery platform. In their study, piezoelectric polyvinylidene-trifluoroethylene (P(VDF-TrFE)) NFs with diameters of approximately 30 nm were synthesized by preparing a solution containing 4.0 wt% P(VDF-TrFE) (70/30 mol%) in a 50:50 weight ratio of DMF and tetrahydrofuran (THF). The solution was supplemented with a 1.5 wt% pyridinium formate buffer and 0.05 wt% BYK-377 to increase its conductivity and decrease the surface tension, respectively. NFs with average fiber diameters of approximately 70, 100, 200, and 500 nm were synthesized separately from solutions of 6.0, 7.0, 11.5, and 17.5 wt% P(VDF-TrFE), respectively. The P(VDF-TrFE) NF membranes were then annealed at 90 °C for 24 h to further improve their piezoelectricity. Crystal violet was chosen as a cationic model drug because of its ease of use in confirming adsorption and quantifying release via colorimetry. The intrinsic negative zeta potential of NFs was used to electrostatically load cationic drug molecules, with surface potential variations triggered by an external mechanical actuation.

Huo et al. [53] developed a NF platform using a blended electrospinning technique to achieve transdermal DDSs. In their investigation, PCL (10% in acetic acid (*w*/*v*)) and collagen (Col) (50% in acetic acid (*w*/*v*)) solutions were blended at various mass ratios, and artemisinin (ART) (10% of total polymers) was dissolved in the combined solution. ART-loaded PCL/Col NFs were prepared via blended electrospinning. Their study found that when the amount of PCL in the PCL/Col NF matrix increased, the hydrophilicity of the NFs decreased but the hydrophobicity increased. Experiments on swelling and weight loss demonstrated the influence of changes in PCL content on the characteristics of PCL/Col NFs. The degree of swelling and weight loss decreased as the PCL concentration in the PCL/Col NF matrix increased. The drug release experiment confirmed the hydrophilic and hydrophobic variations in the PCL/Col NFs, as well as the sustained release behavior. The slow-release impact of ART becomes more promising as the fraction of PCL in the PCL/Col NF matrix increases.

### 2.2. Core-Shell NFs

A core-shell configuration can be used when an electrospun NF contains a medication that may be deactivated in vivo before it can perform its timely function [54]. The medication is contained in the core of the fiber in this strategy, allowing the shell to shield it from the harsh in vivo environment before its biological activity is required [55]. Kharagani et al. [56] reported a study on core-shell NFs with the ability to carry multiple drugs, drawing attention to the development of appropriate DDSs for wound dressings. In their study, biocompatible core-shell NFs were developed as a promising dual drug carrier capable of delivering both water-soluble and organic solvent-soluble drugs simultaneously. A PVA solution containing 11% *w*/*w* was prepared by stirring at 60 °C. The obtained solution was then added with 5 mg/mL diclofenac sodium salt (DSS) and mixed until a white suspension of PVA/DSS was formed. The pH of the viscose suspension was increased to 9 with 0.5 M NaOH to obtain a transparent PVA/DSS solution. The clear PVA/DSS solution was then placed in a 20 mL syringe with a capillary tip gauge of 20 at a distance of 12 cm from the collector and supplied at 17 kV. Electrospinning was continued for 24 h to obtain PVA NFs with a thickness of 0.16 mm. The viscosity of the PVA polymer was measured as 7.51 g/(cm∙s) with a surface tension of 39.85 mN/m and a conductivity of 1.12 ms/cm. The obtained PVA/DSS NF mat was immersed in a crosslinking solution prepared with a mixture of glutaraldehyde (GA):ethanol:hydrochloric acid (2:1:0.2 g) in a 50 mL glass round bottom flask for 10 min, and the volume was adjusted to 50 mL by adding acetone. The unreacted HCl and GA were washed out of the crosslinked PVA/DSS NF mat with deionized water for 5 min and dried at 60 °C for 30 min. Polyacrylonitrile (PAN) solutions of different concentrations (0.5, 1, 2, 3, 4, and 5%) were prepared in a DMF solution with stirring for 12 h, and 5 mg/mL gentamicin sulfate (GENS) was added to the obtained solutions. The PVA/DSS NFs from the previous step were separately immersed in each PAN/GENS solution for 1 min. The PVA/DSS NFs that contained the PAN/GENS solution were immediately immersed in ethanol for 30 s to crosslink the PAN solution. The final product was immersed in deionized water for 10 min to wash out the ethanol and DMF, and the obtained mat was dried at 37 °C for 1 h.

Li et al. [57] fabricated core-shell NFs to achieve sequential release of two drugs (Figure 4). In their study, a local DDS was developed from an emulsion electrospun polymer patch containing hydrophobic 10-hydroxycamptothecin (HCPT) and hydrophilic tea polyphenols (TP) in the shell and core of the NF, respectively. Through emulsion electrospinning, methoxy poly(ethylene glycol)-block-poly(lactide-co-glycolide) (mPEG-b-PLGA) and dextran emulsion with HCPT in the oil phase and TP in the aqueous phase were developed. To reduce the surface tension of the oil phase, 5.0 wt% triethyl benzyl ammonium chloride, based on mPEG-b-PLGA, was used as an emulsifier and added to the PEG-PLGA/CHCl_3_ solution before emulsification. Then, 1.0 mL of an aqueous solution of 45.0 mg∙mL^−1^ dextran was slowly dropped into 11.0 mL of the above oil solution and emulsified at a circulation rate of 6500 r∙min^−1^ for approximately 5 min. The above emulsified solution was filled into a 10 mL syringe, and the electrospinning parameters were set as follows: electrical voltage: 26 kV; inner diameter of the spinneret: 0.4 mm; injection rate of the solution: 1 mL∙h^−1^; inclination angle of the needle with respect to the horizontal: 10°; distance of the air gap: 15 cm. The electrospun fibers were collected on a grounded aluminum sheet.

Wang et al. [58] prepared core-shell NFs for DDSs and tissue engineering because they have latent structures that can encapsulate various drugs and achieve tailored release profiles. Here, blends of poly(L-lactide-co-ε-caprolactone) (PLCL) and gelatin were used to electrospin the shell of the core-shell NFs, which could serve as a protective layer for the controlled release behavior of the model drug DOX. In their study, the core solutions were prepared by dissolving DOX in deionized water and then stirring for 2 h at room temperature to obtain homogeneous solutions. The shell solutions were prepared with 15% (*w*/*v*) PLCL and gelatin in HFIP and stirred for 4 h with a PLCL to gelatin mass ratio of 3:1. The shell and core solutions were injected independently using two different syringe pumps at 1 mL∙h^−1^ and 0.1 mL∙h^−1^, respectively. A core-shell needle with an inner diameter of 0.4 mm and an outer diameter of 1.1 mm was used to electrospin the core-shell NFs. An electrostatic discharge generator was employed to apply a high electric voltage of 20 kV between the collector and the core-shell needle tip to cause stretching of the polymer and beam toward the grounded collector to form a core-shell NF. The distance between the core-shell needle tip and the aluminum foil collector was 20 cm. All electrospinning experiments were performed at room temperature (20 °C) and a relative humidity of 40%. The core-shell needle allowed the polymer in a solvent to inseparably coat an aqueous drug solution as it exited the needle.

Kaviannasab et al. [59] reported the controlled release of an encapsulated drug in the core of core-shell NFs when the shell cannot absorb water, or the shell is made of hydrophobic materials. In their study, different mixed solutions of PVP and PCL with total concentrations of 6% (*w*/*w*) at different PVP:PCL ratios of 100:0, 50:50, 30:70, 10:90, and 0:100 were prepared in chloroform:methanol:DMF (70:15:15) as the solvent system. The polymers were completely dissolved in the solvent using a magnetic stirrer at 23 ± 2 °C for 24 h. The solvent system was used to prepare the polymers. Coaxial electrospinning was conducted at a high voltage of 15 kV and a distance of 18 cm between the nozzle tip and the collector using a trial-and-error method. Core–shell NFs were collected on a rotating cylinder at a speed of 180 rpm. The flow rate of the core was kept constant at 0.2 mL∙h^−1^, while the flow rate of the shell solution was 0.2 or 0.4 mL∙h^−1^. The amount of drug in the polymer solution was 8% by weight of the total polymer dissolved in the polymer solution only 1 h before the electrospinning process. The study showed that the drug was released rapidly when the amount of PVP in the shell was high. With the opening of the surface pores, the access to the core increased, and the release of the shell was sudden. Increasing the water content of the shell allowed rapid access to the core. When the core was made of PVP, the drug was released faster owing to the hydrophilicity of PVP. By controlling the polymer ratio in the shell and modifying the type of polymer in the core, the release rate of 5-fluorouracil (5-Fu) can be adjusted for each type of target tissue based on the diagnosis and treatment protocol.

### 2.3. Layer-by-Layer (LbL) Assembly

LbL is an effective approach for creating prospective biomaterials using multilayer coatings [60]. Following electrospinning, multiple layers on the surface of NFs are created by depositing alternating layers of oppositely charged materials with wash stages in between [61]. Various techniques, including immersion, spin, spray, electromagnetic, and fluidics methods, can be used to accomplish this goal [62]. The LbL adsorption approach provides enormous material freedom and structural design flexibility, which are well matched to the manufacturing demands of drug delivery materials requiring complex designs [63]. Rezk et al. [64] conducted research on bilayered NFs and membranes loaded in wound dressing applications with titanium dioxide (TiO_2_) and tetracycline (TTC). Using the electrospinning device described here, double layers of electrically spun NFs were fabricated. In their investigation, a PCL solution was produced using a chloroform–methanol ratio of 3:1. Polydioxanone (PDO) loaded with TiO_2_ nanoparticles (NPs) was the second layer of NFs. Here, 8 wt% PDO was dissolved in HFIP, and the mixture composite was ultrasonicated for 15 min before electrospinning. PCL–PDO–3% TiO_2_–5% TTC (PP3T5T) and PCL–PDO–5% TiO_2_–5% TTC (PP5T5T) were referred to as two-layer samples. The aim of their work was to develop a new functional two-layer membrane loaded with TiO_2_ and TTC for wound dressing applications. The initial release of bursts followed by a continuous control release of TTC for four days was shown in the study of in vitro drug release. Bilayering of NFs was performed in vitro against Gram-positive and Gram-negative bacteria (Escherichia coli), which are very antibacterial in comparison with PCL NFs. Compared to drug-positive samples, they are highly antibacterial. Their study showed that a bilayered membrane is more compatible with antibacteria and cells than the control fiber. This implies that the manufactured biocompatible fabric is suitable for several wound dressing applications. In their study, the exterior layer of PCL is hydrophobic in nature to preserve its mechanical characteristics and protect it from the external environment. A homogeneous distribution of TTC medication on the NFs appeared in the second layer loaded with TTC. The results show that the NFs loaded with TTC have a significant antibacterial effect on *E. coli* and *S. aureus* from drug release. The fibroblast cell adherence and proliferation of the two-layer membrane are superior to those of the control fibers; therefore, it may be utilized as a possible bond in applications such as controlled administration of drugs and injury healing as a new functional composite NF mat.

Schoeller et al. [65] presented their findings on PLGA electrospun NFs for controlled drug release based on pH-responsive chitosan (CS)/alginate (ALG) polyelectrolyte complexes. Their work shows that the surface of electrospun NFs enables the introduction of more functionality while preserving the membrane characteristics of high porosity and high surface-to-volume ratio. In their study, CS and ALG layers were successively deposited to produce a polyelectrolyte complex through the PLGA NF LbL mount to introduce a pH response for regulated ibuprofen (IBU) release. Spectroscopy and zeta-potential testing indicated the deposition of polysaccharides on the surface of the fibers. Positive surface load (16.2 ± 4.2 mV, pH 3.0) came from polycationic CS, which directly regulate the interaction between the model drug IBU loaded within the polyelectrolyte complex and the LbL coating. The acidic pH release of IBU was decreased by the interaction between the medicine and the coating, compared to the neutral pH. The mesh proposed is a viable option for the development of medicinal products required to circumvent the acidic environment of the digestive tract. A solvent solution of 85/15% (*v*/*v*) of THF/DMF dissolved the PLGA to reach an ultimate polymer concentration of 15% (*v*/*v*). Benzyltriethylammonium chloride was added to the solution at 0.1 % (*w*/*v*). The polymer solution was filled and loaded onto a spring syringe with a diameter of 0.456 mm (21G). The polymer solution was released at 20 μL∙min^−1^ with a potential differential of 20 kV (+15/−5 kV) on a spinning drum (50 rpm) covered with aluminum foil at a needle-to-collector distance of 20 cm. The membranes were electrospun for 4 h and the carpets were then dried overnight in a vacuum oven at 40 °C. An ultrathin plasma cover (approximately 20 nm) was employed to insert an oxygen-containing functional hydrocarbon layer onto the surfaces of the electrospun NFs. To improve adhesion, stability, and superficial function, the covering consists of a vertical chemical gradient, which changes the flow of power and gas during the plasma deposition process. CS and ALG were then used to coat the plasma-coated electrospun NFs. CS was dissolved for 48 h in 0.8% (*v*/*v*) acetic acid for a final concentration of 0.1% (*w*/*v*), and ALG at a concentration of 0.1% (*w*/*v*) was dissolved for 48 h. Using 0.1 M HCl, both solutions were modified to pH 5. Nanofibrous mats were cut into circles with a diameter of 50 mm (100 mg) and placed on a Büchner filter membrane. The mesh under vacuum for each side of the mesh subsequently absorbed 50 mL of the CS solution. Membranes containing 150 mL of Milli-Q water on either side of the net were then cleaned three times to eliminate excess CS from the surface. An identical technique for washing and coating was used for the ALG layer. This method was continued until the required number of layers (5, 9, and 15) was attained. At the end of the process, the membranes were dried in a vacuum oven at 40 °C. Before further characterization, the coated NFs were maintained under vacuum. The meshes with 5, 9, and 15 layers were called (CS/ALG)_5_, (CS/ALG)_9_, and (CS/ALG)_15_, respectively. IBU was utilized as a model drug to examine the kinetics of electrospun loading constructions in response to pH. To load the medication, the mesh (5.4 ± 1.9 mg) of a 2 mL saturated solution of IBU with various numbers of polysaccharide layers was submerged in Milli-Q water (0.021 mg∙mL^−1^) at pH values of 2.0, 5.0, 7.0, and 10.0.

Cheng et al. [66] have published research on regulated co-allocation of growth factors by assembling core-shell NFs LbL to improve bone recovery (Figure 5). The authors have shown that both osteogenic and angiogenic growth factors govern the regeneration of bone tissues in a coordinated event cascade. The goal of their work was to design a dual growth factor release system to promote bone regeneration in a time-controlled manner. They manufactured core-shell nanofibrous SF/PCL/PVA mats by coaxial electrospinning and LbL methods to integrate bone morphogenetic proteins (BMP2) into the nucleus of the NFs and attach them to the surface. Their study confirmed that BMP2 was continuously released, and connective tissue growth factor (CTGF) was promptly released. Experiments both in vitro and in vivo have shown advantages in the recovery of bone tissues using the dual drug release method. In vivo tests showed an improvement of 43% in bone regeneration compared to individual BMP2 release. The time-controlled release offered by the NF-core mounting system is a potential bone-healing approach.

In their study, sequences of NFs were integrated using coaxial electrospinning and the LbL method in three forms: (1) (SF/PCL)/PVA core-shell structure loaded into a BMP2 core with the CTGF fixed onto a surface using LBL ((SF/PCL)1:n/PVA-LBLn), (2) (SF/PCL)/PVA core-shell structure loaded into an ((SF/PCL)1:n/PVA) core, and (3) (LBLn). In the manufacturing of core−shell NFs, SF and PCL were dissolved into an HFIP solution and supplied using a syringe pump to the outside coaxial needle at ratios of 1:2, 1:3, 1:4, 1:5, and 1:6. BMP2 was also integrated into an 8% aqueous PVA (10 μg/mL) solution, and a new syringe pump was used to transfer the BMP2/PVA solution into the inner needle. These two independent pumps regulated the flow rates in the capillaries. The core and shell solution fluid flow rates were 0.6 and 1.8 mL/h, respectively. Between the spinneret and the collector, a DC voltage of 22 kV was applied (aluminum mesh). For the coaxial electrospinning method, the special characteristics of the electrospinning system were the same. All electrospinning operations were carried out at 25 °C and 60% humidity. A CS solution (1 mg/mL, pH = 4.5) and a CTGF solution (10 μg/mL) were selected as the positively charged and negatively charged materials, respectively. Alternatively, the negatively charged (SF/PCL)1:n/PVA mats were blended on the nanofibrous mats for 15 min to form CTGF and CS coatings. The nanofibrous mat was rinsed with NaCl solutions (1 mg/mL) three times after deposition of each layer for 3 min to remove the remaining solution. These layering procedures were repeated until the number of planned layers was attained. LBLn identifies the layers of CS-CTGF bilayers, where n is the number of layers and CTGF is the ultimate layer. Their study demonstrated great improvement in shell formation and bone tissue regeneration by the coordinated temporal release of BMP2 and CTGF using the core−shell nanoparticle montage and LbL technology. Their study also highlights the necessity of temporal regulation for the release of the growth factor in the therapy of bone abnormalities.

## 3. Drug Loading Mechanism

The solubility of the drug in the polymer solution is critical for determining the optimal loading technique, whereas the drug loading technique is critical to the release process. Hydrophobic medicines, such as DOX and paclitaxel (PTX), are generally processed using organic solvents. For example, peptides and proteins are better handled by water-soluble polymers. The various methods for drug loading are discussed in this section (Figure 6).

### 3.1. Encapsulation

The drug blending technique with NFs and polymeric solutions remains the most important among all sophisticated techniques for drug loading and integration with NFs. To prepare encapsulated medicines using a single-phase electrospinning process, the product is distributed or dissolved in a polymer solution [67]. Jannesari et al. [68] researched electrospun NFs for the controlled release of medicines as a new wound dressing matrix (Figure 7). In their research, an affordable and useful mixing approach was utilized to accomplish simultaneous regulated release of ciprofloxacin HCl (CipHCl) and high swelling capacity, as well as wound healing coverage conformity. Poly(vinyl acetate) (PVAc) was mixed with a PVA solution as a low-cost hydrophobic polymer. The CipHCl polymer solution was added 2 h before electrospinning. The addition of drugs decreased the size and reduced the dispersion of electrospun NFs because of the reduction in the viscosity of the solution. The number of NF-loaded drugs influenced the release behavior as well as the first release of NFs from PVA and PVAc. Regardless of the initial release speed, the release rate of PVAc electrospun nanofibrous mats was relatively slow: approximately 50% of the medications were released linearly over eight days, whereas the PVA nanofibrous mats released the drug in three days. Using the PVA:PVAc (50:50) NF mats, the drug release behavior was severely impaired, the quantity of drug released in the early stage decreased, and the drug release profile remained for longer periods compared to that of PVA NF mats. The level of swelling compared to that of PVAc increased considerably. Increasing the density of the mixed NF mats regulated the initial discharge and release rate of drugs as well as enhanced the swelling, making the mats appropriate for deep wound healing. Fickian drug release kinetics were observed in all formulations.

In another study, Cao et al. [69] investigated the ribonucleic acid (RNA) interference by an NF-based siRNA delivery system using an encapsulation method. Although the authors experienced severe processing conditions during electrospinning, the bioactivity of siRNA was preserved for a long time. The nanofibrous scaffold may be used as a continuous release siRNA reservoir to achieve long-term gene therapeutic impact while architectural signals support and directly seeded cells by encapsulating labile siRNA molecules in polymer fibers. The investigators showed that local siRNA availability through scaffolding also increases gene silencing compared to passive siRNA absorption in the absence of a transfection reagent. To create a 14 wt% polymer solution, PCL was dissolved in 2,2,2-trifluoroethanol (TFE). PEG was added to the PCL solution to control the siRNA release. siRNA/ODN in diethylpyrocarbonate treated Tris-EDTA buffer was added to the polymeric solution before electrospinning. For in vitro siRNA transfection efficiency assessment, GAPDH siRNA was used as the siRNA model. The release of siRNA from tissue engineering scaffolds can serve as an essential supplement to local siRNA delivery to provide an appropriate tissue regeneration environment as well as RNA interference modulation of cells. In combination with nanofibrous scaffolds, these biofunctional substrates can provide biochemical signals and seed-cell contact guidance.

Meng et al. [70] reported the use of electrospun PLGA/gelatin NFs as a potential DDS. The authors attempted to develop nanofibrous PLGA/gelatin composite scaffolds as medical carriers with the combined advantages of suitable mechanical properties, bioactive surfaces, and controlled degradability. The polymer composite solution was prepared by dissolving PLGA and gelatin in TFE. Subsequently, fenbufen (FBF) was added into the composite solution. The obtained homogeneous composite solution with complete dissolution of the drug was electrospun. The NFs were shown to have a smooth surface shape and FBF-loaded drug aggregates. The hydrophilicity of PLGA/gelatin scaffolds was improved by increasing the gelatin content, which resulted in increased FBF release. The study showed that the orientation of NFs might impact the FBF release pattern of FBF-loaded nanofibrous PLGA/gelatin scaffolds. Crosslinking with GA vapor caused the drug release rate to decrease. The pH value of the buffer solution may alter the physical state of the polymer matrix, which influences the release rate of the drug; in buffer solutions with different pH values compared to solvent cast films, the nanofibrous scaffold is more stable.

### 3.2. Chemical Immobilization

More applicable strategies include immobilization following chemical fiber surface treatment, which makes the fiber adhesion characteristics adaptable to further immobilization by adding relevant functional groups such as amines, carboxyls, hydroxyls, or thiols. Chemical conjugation techniques are better to finely observe the quantity of the drug integrated into the NF mesh than to capture the drug physically on the surfactant surface. Cho et al. [71] have reported the use of electrospun nanostructures with topographical signals for neuronal differences in mesenchymal stem cells (MSCs) in combination with nerve growth factor (NGF). In their investigation, a nanopattern-based fibrous matrix was chemically combined with the NGF (Figure 8). 

The amine groups were exposed on the surface of the NFs because of the electrospinning of PCL/PCL-PEG containing amine groups. The NGF was chemically modified by conjugation between the carboxylic acids of NGF and the surface-exposed amine groups of the NFs. To investigate the effects of aligned nanotopography and conjugated NGF on the MSC transdifferentiation, MSCs were grown on aligned nanofibrous surfaces. Earlier, an epidermal growth factor was chemically bound on the nanofibrous mesh surface, and keratinocytic differentiation in the epidermal cells at areas of injury was considerably enhanced. Therefore, MSCs were also predicted to transmit to neuron-like cells through the chemically immobilized NGF. To compare the NGF-adsorbed nanofibrous surfaces, the release profile of NGF from the NGF-conjugated matrix was studied.

Im et al. [72] investigated the initial burst of a hydrogel fiber crosslink modified by fluorination prior to adding a model pharmaceutical. In their study, a PVA solution was prepared by dissolution in distilled water and combining with Procion Blue. The NFs were prepared using an electrospinning technique, and afterward, a direct fluorination approach was used to modify their surfaces. Fluorination was performed using highly pure fluorine gas (99.99%). By adjusting the electrospinning conditions, the diameter of the NFs could be regulated to produce thinner or thicker fibers. Thinner fibers provided a greater surface reaction site for fluorination. C–F and C–F_2_ bonds were effectively introduced by increasing the gas concentration during fluorination. Perfluorination at high fluorine pressure resulted in the formation of additional C–F_2_ bonds, making the surface of the fibers hydrophobic and altering the swelling behavior of PVA fibers. When only C–F bonds were produced by fluorination, the initial drug burst was greatly reduced, and the entire release duration was enhanced by approximately 6.7 times.

### 3.3. Physical Adsorption

The physical non-specific random adsorption is regulated by the electrostatic, hydrophobic, hydrogen, van der Waals forces, and fiber–surface interactions. However, non-specific adsorption causes the formation of weak bonds, which can lead to large explosion releases. Therefore, it is infrequently utilized in cancer therapy applications [73]. Surface-modified prefabricated nanofibrous meshes with an exceptionally high surface-area-to-volume ratio can provide drug-friendly physical immobilization on the surface, resulting in greater drug loading quantity per unit mass than any other device [74]. The fast release of pharmaceuticals from the NF surface can allow for easy dose management of some therapeutic agents, making them suitable for some specialized applications such as the prevention of bacterial infection that occurs within a few hours following surgery [75].

Khampieng et al. [76] conducted a study on electrospun doxycycline hyclate (doxycycline hydrochloride hemiethanol hemihydrate; DOXY-h)-loaded poly(acrylic acid) (PAA) NFs for in vitro drug release and antibacterial properties based on physical adsorption. Their primary goal was to create thermally crosslinked PAA NFs containing DOXY-h, which is a broad-range antibiotic. Drug absorption on NFs was achieved using the Freundlich model, which involved mass transport of an adsorbate from the solution phase. The electrospinning solution was prepared by dissolving PAA in ethanol by mechanical stirring until it became homogeneous. Subsequently, ethylene glycol, a crosslinking agent, was added to the PAA solution. Before the electrospinning process, H_2_SO_4_ was added to the electrospinning solution to catalyze the thermal crosslinking of the PAA NF mats. To load DOXY-h into the crosslinked PAA NF mats, drug sorption was chosen because it is simple and practical, and it avoids DOXY-h degradation during the electrospinning and thermo-crosslinking of the PAA NFs. By soaking crosslinked PAA NFs in a DOXY-h aqueous solution at concentrations of 125, 250, 500, and 1000 μg/mL, four types of PAA/DOXY-h NF mats were produced: PAA/DOXY-h125, PAA/DOXY-h250, PAA/DOXY-h500, and PAA/DOXY-h1000. The DOXY-h adsorption isotherms were typical of the Freundlich model. Under physiological conditions, the release of DOXY-h from PAA/DOXY-h NF mats revealed an initial burst release characteristic driven by the Fickian diffusion process. These NFs were shown to be more efficient against Gram-positive bacteria such as *S. aureus* and *S. agalactiae* than against Gram-negative bacteria such as *P. aeruginosa*. 

Chen et al. [77] reported physical adsorption-based drug loading and release using PLA NFs. In their study, PLA NFs were electrospun, and anticancer drug daunorubicin was accumulated on PLA NFs mixed with TiO_2_ NPs. The relative surface of TiO_2_ NPs and PLA polymer NFs in pH 7 aqueous solutions was negatively charged, whereas daunorubicin was positively charged. Because of electrostatic contact and other non-covalent interactions, daunorubicin could easily self-assemble onto the surface of TiO_2_ NPs and PLA polymer NFs. Based on the drug accumulation behavior of the broad-spectrum anticancer agent daunorubicin on PLA NFs, the blends of nano-TiO_2_ and PLA NFs could provide promising scaffolding for non-covalent attachment of multiple drugs, which could efficiently enhance drug uptake in target leukemia cells. The daunorubicin solution and other aqueous solutions (pH 7.0) were prepared using ultrapure water, which was distilled and subsequently filtered at 25 °C. PLA NFs and TiO_2_ NPs were suspended in ultrapure water and maintained at 4 °C. The study found that these nanocomposites could easily stimulate the anticancer drug daunorubicin to accumulate in leukemia K562 cells, resulting in significantly increased intracellular fluorescence intensity when the nano-TiO_2_ and PLA NFs were combined with daunorubicin. The rationale behind this is that much more daunorubicin could readily accumulate in target cancer cells because of the efficient self-assembly of anticancer drug molecules on the surface of nano-TiO_2_ and PLA NFs, which can effectively facilitate the specific association with the NFs and biologically active molecules.

In another study, Volpato et al. [78] described the physical adsorption of heparin-containing polyelectrolyte complex NPs (PCNs) and a basic fibroblast growth factor (FGF) on CS NFs (Figure 9). The explosion was avoided by modifying the fiber surface after manufacturing by adding a single bilayer of N,N,N-trimethyl CS (TMC) and heparin multilayer polyelectrolyte. The utilization of high-porosity CS-based NFs and a wide area for the binding, stabilization, and controlled release of heparin-binding development factors was proven. CS is a polysaccharide that contains amine and is a weak polycation that is protonated in acidic media. In their work, the authors created networks of CS NFs, which could appear on the surface and release NPs that contain growth factors. FGF-2/PCN complexes were electrostatically adsorbed onto the CS fibers. CS-heparin PCNs, which include FGF-2 as the model heparin-binding growth factor, were transformed into CS-heparin networks. FGF-2 was shown to be free up to 30 days in zero order from the fibers. FGF-2 release was regulated by further complexation of the polysaccharides of the adsorbed NPs. When the fibers were further modified by a single bilayer of TMC and heparin polysaccharide-based polyelectrolyte multilayer (PEM), the release of the FGF-2/PCN complexes was inhibited throughout the testing. The mitogenic activity of the FGF-2/PCN complexes was also assessed in relation to the proliferation of ovine bone marrow-derived MSCs. FGF-2 activity was preserved over 30 days with complexity. Their study verified its bioactivity by encouraging the proliferation of MSCs using both FGF-2/PCN solution complexes and adsorption onto PEMs by biological assessment of the FGF-2 activity. Because of the cell convergence in some of the wells, there were major ambiguities regarding the cell proliferation data of the FGF-2/PCN solution complexes, which started lifting the cells.

Ma et al. [79] worked on a porous NF device constructed of a CS/polyethylene oxide (PEO) combination, which is an adaptable technique to modify the surface based on electrostatic interactions. After manufacturing the PEO NFs, they were soaked in water and loaded into a solution using an anticancer medication. In their study, the authors reported beneficial combinations, including well-mixed CS and hyaluronic acid (HA), in developing and characterizing natural–natural polymer NFs. The initial contribution of their work was the onsite production of NFs and pores containing PTX NPs. The second was the finding that the positively loaded CS-NFs and negative polymer HA did not have any detrimental effects or reactions because of cross-connection or photopolymerization with GA. A systematic investigation was carried out on the shape of pores, drug-loaded NFs, and PTX releases. The findings of cell cultivation indicated that the NF mat was effective in preventing and proliferating cell adhesion. The interaction between the positively charged porous polymer CS and the negatively charged polymer HA was demonstrated in their work.

## 4. Stimuli-Responsive on-Demand Drug Release

It is generally acknowledged that medications must be delivered in a regulated manner to target areas to increase therapeutic efficacy while reducing or avoiding unwanted effects [49]. Stimuli-based DDSs have demonstrated tremendous potential for the effective targeting of active pharmacological moieties [24]. Combining electrospinning methods with stimuli-sensitive materials to create stimuli-responsive drug-loaded NFs is an intriguingly growing topic [28]. A few stimuli-responsive systems have been integrated into electrospun nanofibrous delivery vehicles by direct co-electrospinning or post-modification via chemical or supramolecular linkages, such as crosslinkers, or on the polymer backbone [80]. Stimulation causes a volume change or disassembly of the delivery vehicles, which results in pulsatile drug release [30].

### 4.1. Temperature

Temperature-responsive drug-loaded electrospun NFs are produced from polymers with abruptly varying solubilities [37,81]. This is based on the competition between hydrophilic and hydrophobic molecules on the polymer chain [82,83]. The human body temperature has a restricted range and fluctuates during the day. Fever, hyperthermia, and hypothermia are all examples of deviations from normothermia [51,84]. Fever is a sign of a variety of medical conditions, including infectious diseases, immunological diseases, cancer, and metabolic imbalances [49,85,86]. External sources can be used to either heat or cool tissues, resulting in localized hyperthermia or hypothermia. This implies that temperature may be controlled and utilized as a trigger to modify drug release [80,87]. Tran et al. [88] reported a study on controllable and switchable drug delivery of IBU from temperature-responsive NFs. Electrospun NFs of PNIPAM and hydrophobic PCL polymers were used without burst effects at both room temperature and at temperatures beyond its lower critical solution temperature (LCST) for controlled and variable IBU release. These NFs may be used for transdermal drug administration, which greatly improves the effectiveness of drug dependence and drug misuse. Temperature has a negligible impact on the IBU diffusion rates from PCL/IBU NFs. For PNIPAM/IBU NFs, a considerable burst effect is achieved at 22 °C, but at higher temperatures, the rate of diffusion and the burst effect are significantly decreased. The explosive impact is considerably decreased at both 22 °C and 34 °C for the PNIPAM/IBU/PCL NFs. At 22 °C, the diffusion rate is 75% higher than that at 34 °C. Of course, there are several practical applications in the pharmaceutical and medical sciences, and such a controlled and switchable delivery system can readily be found. 

Kim et al. [89] have been investigating an on–off switchable dextran release temperature sensitive electrospun NF. In their study, thermal air curing was performed for post-crosslinking of the NIPAAm NFs. The first step was polymerization and electrospinning of thermo-crosslinked poly(NIPAAm-co-N-hydroxymethylacrylamide; HMAAm) into the NFs. HMAAm was selected because its group of methylols may be chemically interconnected with self-containment when heated. The NFs were then crosslinked at 110 °C by heat treatment. In response to cycles of temperature alternation, the resultant crosslinked NFs displayed fast and reversible size changes in aqueous environments, but their fibrous shape was maintained even during the temperature alternation cycles. The copolymers blended with fluorescein isothiocyanate (FITC), and controllable release of FITC-dextran from crosslinked NFs in reaction to external temperature alternation cycles was achieved. For drug release, the NFs were incubated at different temperatures in a PBS solution.

A study regarding on-demand drug release from custom blended electrospun NFs has been published by Amarjargal et al. [82]. T*_g_*-based drug release from polymeric NFs is directly linked to drug diffusion. Heating over the T*_g_* value facilitates transition to the rubbery state in NFs. This allows considerable drug diffusion by increasing the mobility of the polymer chains, resulting in controlled drug release. The authors intended to build an effective system of drugs based on a combination of non-cytotoxic polymers that are used in the medical sector. Its adjusted T*_g_* prevents active molecules from being released prematurely at physiological temperatures. Their research utilized RhB for the development of on-demand releases blended by adjusting the ratio of Eudragit^®^ RS100 (ERS), a biocompatible copolymer to NFs of bioinert poly(methylmethacrylate) (PMMA). The study revealed that the use of electrospinning technology effectively resulted in a simple and economical approach for manufacturing a novel class of easily processed cytocompatible, thermo-sensitive ERS/PMMA membranes. In vitro investigations of drug release revealed that a suitable T*_g_* is produced, changing the relationship between polymers and resulting in a combination of physiologically optimized drug and thermal releases. The produced NFs provide a viable substrate for implanted drug delivery with pulsatile delivery devices.

Another study using T*_g_*-based controlled drug release was conducted by Pan et al. [90]. In their investigation, T*_g_*-modulated NFs were activated for greater antibacterial release at a physiological temperature of 37 °C (Figure 10a). It has not been used to achieve thermo-activated drug release from NFs to prevent bacterial infections during wound healing. The authors initially created electrospun NFs from ERS and bioinert PMMA blended polymers to produce a tailored wet T*_g_*, or thermal stimulation. A model drug, octenidine (OCT), was then used and integrated into the mixed polymer. By adjusting the ERS/PMMA ratio, regulated OCT release at physiological temperature was achieved at an optimal wet T*_g_* of the NFs (Figure 10b). Because of the controlled OCT release regulated by the thermal switch, the produced nanofibrous membrane demonstrated excellent antibacterial activity against both Gram-positive and Gram-negative microorganisms at physiological temperatures. The observations and conclusions presented here are not only scientifically intriguing but can serve as bases for noninvasive self-stimulated release of antimicrobials for the treatment of skin wound infections, hence reducing antibiotic misuse. T*_g_*-based drug release aims to prevent antibiotic misuse by providing regulated release at or above the physiological temperature.

### 4.2. Light

Light-responsive materials manifest great potential for providing distant and precise operation that may be readily directed into specific regions of therapeutic applications [80]. The photoresponsivity of these materials is frequently based on photoisomerization of component molecules, which undergo substantial conformational shifts between two states in response to light absorption at two distinct wavelengths [49].

A versatile platform based on electrospun NFs for NIR light-driven biomedical use was reported by Nakielski et al. [50]. The authors successfully manufactured a platform with biomimetic structural characteristics for the on-demand distribution of medicines. This platform consists of electrospun PLLA NFs loaded with a drug model RhB encapsulating P(NIPAAm-co-NIPMAAm)/GNR plasmonic hydrogel. The study outlines the discovery of a simple technique to substantially increase the medication supply to a specified tissue. Cascades, such as NIR light, absorbed by GNRs into heat, are used to regulate the release. These stimuli, in turn, induce hydrogel structural changes and accelerate the kinetics of drug release. Analysis at different temperatures were performed to examine the reactivity of the cushion platform. The same method may be used for non-healing infected wounds when a laser pillow can release antimicrobial products locally, while the heat generated can lessen bacterial colonies. Because the platform effectively satisfies the biocompatibility and stability requirements for thermoresponsive nanomaterials, the PLLA/P(NIPAAm-co-NipmaAm)/GNR system is a good choice for achieving the on-demand release of medications in conjunction with photothermal processing. In another study presented by our group, PNIPAM NFs containing GNRs and drugs that can be controlled by NIR light irradiation were prepared (Figure 11) [80]. Stable PNIPAM NFs were generated by a crosslinking process with OpePOSS to prevent them from dissolving in water below the LCST. The thermal/optical responsiveness of PNIPAM NFs containing GNRs and the drugs produced by electrospinning was high. The results demonstrated that the NFs were structurally stable and had a very large surface-area-to-volume ratio for successful on-demand drug delivery. The introduction of GNRs into NFs resulted in a substantial photothermal impact. Because of the thermal sensitivity of PNIPAM, the heat generated by the GNRs during NIR light irradiation might govern the swelling and deswelling behavior of the NFs, resulting in drug release. The biocompatibility of the NFs was validated through cell studies using camptothecin (CPT) as an anticancer drug.

A study by Altinbasak et al. [91] on a photothermally induced on-demand antibiotic release platform examined a polymer-embedded reduced graphene oxide (rGO). A NF-based on-demand antibiotic release platform was manufactured using rGO-containing polymer NFs. Crosslinked hydrophilic NFs produced by electrospinning a combination of PAA and rGO exhibited excellent aqueous stability (Figure 12a). Two antibiotics, ampicillin and cefepime, were loaded onto the rGO-embedded hydrophilic NFs fabricated by electrospinning. The PAA NFs showed relatively little photothermal heating; however, PAA@rGO NFs with continuous NIR irradiation resulted in a temperature increase of approximately 67 ± 2 °C. Even reducing the laser power density resulted in a significant increase in the surface temperature of the PAA@rGO NFs in a moist environment, reaching saturation at approximately 51 ± 2 °C within 5 min. Exposure to NIR radiation leads to a release of adequate quantities of Gram-positive and Gram-negative bacteria, whereas negligible antibiotic release is detected under physiological conditions (Figure 12b). There is a strong connection between the antibiotic release caused by the NIR and bactericidal action. The easy manufacturing and modular characteristics of the platform is anticipated to be modified to supply numerous medicines on request for treating various diseases.

Sutka et al. [92] proposed a visible light-sensitive DDS using a coaxial electrospinning technology to produce complex core-shell composite NFs with a PVA shell and PVP core. Goethite (α-FeOOH) serves as the light-visible substance of the drug carrier in the core. In their work, core-shell NFs of (PVA/PVP/α-FeOOH-MB) were prepared. Because of electrostatic interactions, the model drug methylene blue (MB) was adsorbed on α-FeOOH and desorbed during visible light radiation as a result of local heating of α-FeOOH. The results show that the developed materials may be utilized as a visible light-driven medication supply system. In another report, Huang et al. [93] described electrospun upconverting nanofibrous hybrids for wound dressing with smart NIR light-controlled drug release. The authors developed a promising technique for creating an NIR-initiated DDS using heterogeneous PVA electrospun NFs embedded with a photocleavable polymer prodrug and water-dispersible lanthanide-doped upconverting NPs (UCNPs). A well-defined multistep synthesis enabled the production of levofloxacin-PEG with 45% levofloxacin conjugation efficiency to PEG chains via o-nitrobenzyl (ONB) linkages. UCNPs are excellent media for converting NIR to UV light, hence reducing the damaging effects of UV light on tissues. When excited UCNPs are exposed to NIR light, they produce UV light at 365 nm, which can break the ONB bond of the levofloxacin conjugates in the PVA fiber, resulting in controlled drug release. The fibers have a high swelling ratio, which can aid in the evacuation of excess exudates from wounds. Notably, no leakage was detected in the dark. Controlling the release allows the required medication to be delivered at specified times, locations, or as required, without disrupting the dressing. 

Tiwari et al. [94] examined the use of polydopamine (PDA)-based implanted multifunctional NFs for extremely effective photothermal chemotherapy. The authors reported the effective development of a localized anticancer drug delivery platform with bimodal functionality. PCL-DOX fibers were prepared by simple electrospinning, and the surface was modified by chemical polymerization of PDA at different concentrations. PDA has multifunctional properties, such as pH and NIR responsiveness. PDA-coated PCL-DOX mats demonstrated pH and NIR dual responsive behavior, resulting in improved drug release in an acidic medium relative to physiological pH conditions (pH 7.4), which was further enhanced by NIR exposure. The enhanced dissociation of electrostatically bound DOX molecules from PDA, caused by PDA protonation, was linked to drug release. The drug release during NIR irradiation might be attributable to an increase in the local temperature due to the heating impact of PDA. As a result, rising temperature can cause the PCL layer to destroy the drug molecules, reducing its drug retention capacity and weakening the drug–PCL interaction. The thin coating of PCL-DOX membranes had additional nodules on the fiber surface similar to PDA particles. The resulting membranes demonstrated outstanding photothermal behavior and stability in response to an 808 nm NIR laser. Similarly, PDA-coated NFs showed enhanced DOX release at pH 5.5 compared to higher pH solutions (6.8 and 7.4). Furthermore, when PDA-coated fibers were stimulated with NIR light (808 nm, 1.5 W/cm^2^, 5 min), the quantity of DOX released in the local environment indicated on-demand delivery of cancer drugs. The superior toxicity of PDA-covered membranes in response to NIR illumination compared to non-irradiated mats might be due to the synergistic impact of NIR-driven photothermal treatment and the simultaneous increase in DOX release. Their study presented a potential composite material with adjustable drug release characteristics that can be employed in localized treatment of cancer and other diseases.

Obiweluozor et al. [95] described a rapid bioresorbable smart NF device incorporating NIR lethal PDA NPs for efficient chemo-photothermal cancer therapy. The authors reported an effective combination of chemotherapy and photothermal therapy integrated into a single platform for synergistic anticancer treatment utilizing PDO NFs containing PDA NPs and bortezomib (BTZ). It was shown that PDA NPs could be encapsulated in PDO NFs without morphological deformation. Furthermore, when exposed to 808 nm NIR light, the PDO NFs could generate efficient and repeatable heat without loss of heating capacity. The NIR-induced PDO NFs successfully converted light into high thermal energy with great heating capabilities. The NIR-triggered exothermy of PDO NFs might entail optically driven resonance or quantum confinement generated by PD NPs, resulting in fast heating comparable to photo-induced metal NPs. The results confirmed that BTZ-containing PDO NFs permit cancer cell attachment, which results in effective ablation/elimination of cancer cells after 3 min of NIR irradiation. Their research raises the possibility of using drug-loaded NFs in a local combination of phototherapy and chemotherapy against CT26 colon cancer cells. Because this platform can degrade in vivo, long-term adverse effects on normal tissues and organs can be avoided.

### 4.3. pH

Acid-base homeostasis regulates the pH of the human body, maintaining the arterial blood pH between 7.38 and 7.42. On the other hand, many tissues or cell compartments have their own specific pH conditions for optimal functioning [49]. The acidic environment prevalent in tumor tissues can be used to selectively focus the release of anticancer medications at the tumor in response to pH variations via the use of pH-sensitive nano-formulations [96]. Treatment can begin when NFs adorned with targeting modules are manufactured using the electrospinning technique and filled with anticancer medication [97]. When the therapy is initiated, an internal stimulation, such as a low pH in tumor tissues, induces the release of the medication at the exact site of the tumor, allowing the treatment to be applied to tumor cells [98].

Arafat et al. [96] examined the pH-responsive color change of PVA/PAA-based electrospun NFs with bromothymol blue (BTB) and on-demand ciprofloxacin release characteristics. The possibility of pH-responsive electrospun NFs with dual functionality of color change and on-demand drug delivery was demonstrated in their work. When BTB and ciprofloxacin were introduced into the PVA/PAA matrix, the average NF diameter decreased from 222 ± 60 nm to 141 ± 46 nm. There was a slight increase in the average fiber diameter following heat crosslinking, but it was determined to be statistically insignificant. The results showed that there were few new bonds formed following the combination of BTB and ciprofloxacin, resulting in a related peak intensity. A quantitative color change analysis of nanofibrous mats in simulated wound settings demonstrates that there is sufficient color change to identify distinct wound states. Ciprofloxacin is released on demand in several simulated wound situations. Owing to the repulsion of negative -COO- ions, the proportion of ciprofloxacin released nearly doubled after a few hours under simulated wound conditions (high pH release medium). At pH 7 and 8.5, the swelling ratio of the nanofibrous mats increased to 1378 and 1565%, respectively. A high swelling ratio may be beneficial for absorbing wound exudates. An antibacterial study of the samples provided sufficient evidence that the fabricated NFs were efficiently active against both Gram-positive and Gram-negative bacteria (Figure 13). From the successful pH-responsive color change and drug delivery, it can be anticipated that the developed dual-functional pH-responsive electrospun NFs are potential candidates for various wound dressing applications.

Demirci et al. [99] explored pH-responsive NFs with regulated drug release characteristics. pH-responsive poly(4-vinylbenzoic acid-co-(ar-vinylbenzyl)trimethylammonium chloride) (poly(VBA-co-VBTAC)) NFs encapsulating ciprofloxacin were effectively produced in their study using electrospinning procedures for controlled drug release systems. Poly(VBA-co-VBTAC) contains cationic VBTAC units and pH-responsive VBA units. The observed drug release was mostly attributable to drug diffusion or penetration via the polymer matrix because the duration of the experiment (720 min) was insufficient to show polymer degradation. Ciprofloxacin was homogeneously distributed throughout the poly(VBA-co-VBTAC) NFs without producing phase-separated crystalline aggregates. The investigation revealed that the poly(VBA-co-VBTAC)/ciprofloxacin NFs were capable of releasing ciprofloxacin in a regulated manner over an extended period depending on the pH. Because of the increased intermolecular and/or intramolecular contacts, the first burst release increased with increasing pH levels. However, the overall amount of ciprofloxacin released from NFs was greater in the acetate buffer solution than at higher pH levels. These pH-sensitive poly(VBA-co-VBTAC) NFs may lead to the development of novel responsive materials for a variety of biomedical applications.

He et al. [100] claimed that smart cellulose NFs with high biocompatibility allow for sustained antibacterial and drug release via a pH-responsive mechanism. The matrix NFs produced from bagasse pulp cellulose fibers and polyethylenimine (PEI) were utilized as pH-responsive functional reagents to produced new nanosized biomass-based pH-responsive cellulose NF–PEI with high biocompatibility. Cellulose NF–PEI containing DOX has an excellent pH response to convert wettability qualities because it has hydrophilic and underwater superoleophobic properties under acidic conditions and superoleophilic and hydrophobic capabilities under alkaline conditions. After five cycles of switching between an acidic and alkaline media, the pH-responsive wettability of the cellulose NF–PEI surface remained satisfactory. The results showed that amino groups directly affect the surface roughness of the material in response to pH through protonation–deprotonation, and the reversible transformation of pH-responsive wettability on the surface of cellulose NF–PEI was achieved. Cellulose NF–PEI exhibits strong pH-responsive antibacterial activity. 

Mamidi et al. [97] described the development and testing of forcespun functionalized carbon nano-onion-reinforced PCL composite NFs for pH-responsive drug release. The pH-responsive PCL/mercaptophenyl methacrylate-functionalized carbon nano-onion ((PCL/f-CNO) /f-CNO) composite NFs encapsulating DOX were effectively produced using Forcespinning^®^ technology for controlled drug release. The PCL/DOX/f-CNO composite NFs were homogeneous and free from beads. DOX encapsulation had no effect on the morphology of the PCL/DOX/f-CNO NFs. The study indicated that the PCL/DOX/f-CNO composite NFs could efficiently release DOX over time. The interactions between f-CNOs and DOX, as well as electrostatic or hydrogen bonding interactions between f-CNOs and PCL, all contributed to the long-term sustained release of DOX. In vitro degradation studies indicated that even after 25 weeks, the PCL/DOX/f-CNO composite fibers retained their original structure. Wettability observations revealed that the addition of f-CNOs enhanced the hydrophobicity of the PCL/DOX/f-CNO fibers. The presence of f-CNOs improved the mechanical characteristics and biocompatibility of PCL. Over a 15-day period, PCL/f-CNO composite fibers demonstrated pH-responsive DOX release: pH 6.5 showed 87%, while pH 5.0 indicated approximately 99% of DOX release. Cell viability on the surface of the PCL/DOX/f-CNO composite fibers was shown to be excellent in cytotoxicity experiments. Nonetheless, the currently available DOX-loaded composite fibers may be used as biomaterials for pH-sensitive controlled drug delivery.

Jiang et al. [98] described the surface functionalization of electrospun NFs with mussel-inspired proteins for pH-responsive drug delivery. The goal of their research was to develop pH-responsive drug delivery devices using PDA-coated electrospun NFs. It was proposed that the release of charged molecules from air plasma-treated electrospun PCL NFs can be pH-dependent. Moreover, it was suggested that the extra PDA coating might customize the pH-responsive release of charged molecules. The pH of the solution influenced the absorption and release rates of both rhodamine 6G hydrochloride (R6G) and DOX. DOX and R6G are both hydrochloride salts that carry positive charges after being dissolved in water. A negatively charged surface may electrostatically attract such drugs. PDA-coated PCL NFs might trap cations because they have a high surface-area-to-volume ratio and a significant number of hydroxyl groups because of the PDA coating on the surface. When the pH was reduced, the hydroxyl groups on the surface tended to protonate. As a result, the negative charges on the surface of the fibers dribbled away, perhaps causing the release of DOX and R6G from the PDA-coated PCL NFs. A DOX-releasing cell culture medium was used in a 3-(4,5-dimethylthiazol-2-yl)-2,5-diphenyltetrazolium bromide test of H1299, demonstrating that DOX-releasing media could kill more cancer cells at low pH values than at high pH values. This showed that PDA-coated PCL NFs have potential use in oral and topical medication delivery to sites where pH values vary. The work revealed that a mussel-inspired protein PDA covering can precisely adjust pH-responsive loading kinetics and charged molecule release. These new formulations have the potential to be used in drug delivery to particular targets that are affected by pH changes. 

Zhang et al. [101] explored pH-responsive nanocomposite fibers for magnetic resonance imaging (MRI) drug release monitoring. In their study, composite pH-sensitive NFs were produced via electrospinning. PVP-superparamagnetic iron oxide NPs (SPIONs) (a negative MRI contrast agent) and carmofur (a model drug) were mixed into pH-responsive and biocompatible Eudragit polymer fibers (Figure 14). Fibers with smooth cylindrical morphologies were produced with an amorphous dispersion of carmofur. SPION encapsulation in fibers resulted in excellent digestion protection in the acidic environment of the stomach, and in vitro drug release tests indicated fast release of carmofur at pH levels characteristic of the small intestine and colon. Based on these findings, this platform appears to be a promising oral delivery method for colonic cancer. The fibers also show pH-responsive relaxation behavior in the physiological pH range, making them ideal candidates for the development of ultra-sensitive reporters to detect aberrant microenvironments in the small intestine and colon. Chemotherapeutic release and absorption can be significantly influenced by changing the local environment and colonic residence time, making it difficult to provide effective and safe doses. The dynamic process of matrix dissolution/swelling allows water molecules to access the SPIONs, increasing diffusive water access, and thus improving their relaxation rates and relaxivities. The obtained r_2_ relaxivity profiles can be used to determine whether an MRI signal is suitable for monitoring NF dissolution/swelling, and thus the release of carmofur and SPIONs. Because most chemotherapeutic drugs are cytotoxic and nonspecific, their safety remains a major concern, and these formulations may pave the way for a novel approach to significantly reduce off-target adverse effects in chemotherapy.

### 4.4. Electric and Magnetic Field

Electrical fields can cause redox reactions and, in some circumstances, ionization, which can lead to swelling, shrinking, or bending of polymeric drug carriers [37]. Yun et al. [102] created an electro-responsive drug carrier by electrospinning PVA/PAA/multiwalled carbon nanotubes (PVA/PAA/MWCNTs) (Figure 15). MWCNTs were used to enhance the conductivity of the DDS. The swelling ratio of the electrospun NFs increased with the increase in electric voltage. The carboxylic acid groups in the polymer were ionized because of the applied electric voltage. The ionization of carboxylic acid groups caused electrostatic repulsion, which resulted in fiber swelling. Therefore, increasing the applied electric voltage leads to quicker drug release from the electrospun scaffold. 

Kim et al. [51] showed the property of magnetic field responsive switchable drug release for cancer treatment. In their study, NFs were produced from a thermally responsive poly(NIPAAm-co-HMAAm) polymer for the on–off controlled release of dextran via the electrospinning method. Temperature-responsive polymers and MNPs serve as triggers for drug release and sources of heat. The MNPs were incorporated into the NFs. Approximately 50% of the HMAAm methylol groups in the polymer were thermally crosslinked without affecting the shape of the fibers. The LCSTs of the manufactured NFs were around the temperature of the human body. In response to cycles of temperature alternation during the LCST, the crosslinked NFs showed fast swelling and shrinking. Dextran-loaded NFs were manufactured, and the switchable on–off release of dextran from the NFs was observed with a continuous and smooth fibrous structure. The switchable drug release achieved switchable changes in the swelling ratio of NFs in response to alternating on–off switches of the AMF because the self-generated heat from the incorporated MNPs induces deswelling of polymer networks in the NFs. The reported integration of intelligent characteristics into the NFs is attributed to their exceptional surface area and porosity and should be an easy platform for drug administration. Samadzadeh et al. [103] discovered that all magnetic NFs (MNFs) had heat production properties and on–off switchable heating capabilities (Figure 16).

The swelling ratio with reversible changes and the matching drug discharge in response to AMF application with on–off switching were also exhibited. The NFs were fabricated by electrospinning a temperature-responsive copolymer of NIPAAm and N-hydroxymethylacrylamide (HMAAm) [poly(NIPAAm-coHMAAm)] blended with MNPs and metformin (MET)-loaded mesoporous silica NPs (MSNs). The MNPs act as heating sources in response to the AMF. The MNPs in combination with NFs caused local hyperthermia, which increased the drug release. Furthermore, MSNs can release large amounts of a drug gradually, consistently, and precisely because of their large pore volume and surface engineering properties.

### 4.5. Multistimuli

To increase the broad tunability over drug administration, multistimuli-responsive electrospun NFs that respond to a combination of two or more signals have been produced [49,104]. These integrated reactions may occur either concurrently or sequentially. Dual stimuli-responsive drug-loaded electrospun NFs, for example, can trigger the release of medications to an infection site anytime the local pH or temperature deviates from normal [105,106]. Multistimuli-responsive electrospun NFs can be composed of a few single stimuli-responsive electrospun fibers or of macromolecules, polymer mixtures/blends, or surface coatings that respond to several stimuli [107].

Shi et al. [108] investigated the pH and electro-response properties of bacterial cellulose NF/sodium ALG (SA) hybrid hydrogels for dual-controlled drug delivery. Hybrid hydrogels of bacterial cellulose NFs and SA (nf-BC/SA) were produced for DDSs. The stimuli-responsive swelling characteristics and stimuli-responsive drug release behaviors of the nf-BC/SA hydrogels were studied in vitro using IBU as a model drug. With the inclusion of BC NFs, the microstructure of the nf-BC/SA hybrid hydrogels became more stable and accurate. In comparison to pure SA hydrogels, the swelling characteristics of the nf-BC/SA hybrid hydrogels not only preserved pH responsiveness but also effectively increased electro-responsiveness. ALG is prepared as a component of a stimuli-responsive delivery system because of the large number of ionizable -COO- groups in the hydrogel. The swelling ratio increased, which could be attributed to the presence of more -COO- groups in the nf-BC/SA33 matrix. The semi-IPN structure of nf-BC/SA33 provided a larger specific surface area, resulting in a greater number of active -COO- groups, and thus improving electric response sensitivity. Under acidic conditions (pH = 1.5), the swelling ratio was less than 8 times its dry weight, and it rose to more than 13 times when the pH value increased to 11.8, which might be related to SA ionization at higher pH conditions. The swelling ratio of the hybrid hydrogels increased from 8 to 14 times its dry weight as the applied voltage (0–0.5 V) increased. The drug release rate of the nf-BC/SA hybrid hydrogels in vitro was affected by the pH-responsive swelling behaviors, that is, rapid in neutral or alkaline media but sluggish in acidic media. Compared to passive release, drug release may be accelerated by electric stimulation. The rate of release is affected by the applied electric strength.

Yuan et al. [107] explored drug release regulation using pH- and temperature-responsive electrospun CTS-g-PNIPAAm/poly(ethylene oxide) hydrogel NFs. In their study, an EDC- and NHS-mediated coupling reaction was used to successfully synthesize a stimuli-responsive graft copolymer based on CTS and PNIPAAm. The CTS-g-PNIPAAm/PEO blend was electrospun into NFs as a pH- and thermo-responsive hydrogel carrier for regulating the release behavior of bovine serum albumin (BSA). The CTS-g-PNIPAAm/PEO NFs exhibited pH and temperature-dependent swelling behavior, making them suitable for stimuli-responsive drug administration. The drug-release research revealed that CTS-g-PNIPAAm/PEO hydrogel NFs offered a regulated release of the entrapped protein, and the release behavior was controllable by altering the pH and temperature of the medium. Cell proliferation experiments and morphological observations revealed that the CTS-g-PNIPAAm/PEO hydrogel NFs were not cytotoxic. These findings suggest that the current CTS-g-PNIPAAm/PEO hydrogel NFs, as intelligent nanomaterials, have potential applications in regulated drug delivery and tissue engineering. 

Tiwari et al. [94] present a significant step forward in the development of a therapeutic model for cancer treatment by utilizing the pH and NIR dual responsive property of PDA alone in a fibrous mat (Figure 17). PDA coated PCL-DOX mats demonstrated pH and NIR dual responsive behavior in their study, exhibiting improved drug release in an acidic medium compared to physiological pH conditions (pH 7.4), which is further increased by NIR exposure.

## 5. Modes of Drug Administration

The potential of NFs in providing different administration routes, as well as the accompanying obstacles, has been discussed, including NF commercial goods for biomedical purposes. Here, we discuss the capacity of NFs to transport therapies through multiple pathways and their potential to deliver a wide range of treatments for treating diverse illnesses [109,110,111].

### 5.1. Oral Drug Delivery

Because of its noninvasive nature, convenience of use, and increased patient compliance, oral drug delivery is one of the favored methods for drug administration [112]. Furthermore, oral formulations can be developed in a variety of ways, and they can be manufactured inexpensively. In comparison to other methods, the orally administered dose form requires no skill, and is, thus, helpful for chronic conditions requiring regular dosage consumption [109]. NFs can be administered using fast-dissolving drug delivery methods because they can dissolve/disintegrate quickly, allowing the desired drug to be delivered without the need for swallowing or water. Orodispersible medicines dissolve or disintegrate quickly in the mouth without the need for water to facilitate swallowing [113]. Celebioglu et al. [112] described a quick dissolving oral DDS based on electrospinning of cyclodextrin/IBU inclusion complex NFs. The authors used the electrospinning approach to create rapidly dissolving nanofibrous webs from hydroxypropyl-beta-cyclodextrin (HP*β*CyD)/IBU inclusion complexes without the need for any polymeric additive (Figure 18).

IBU was complexed with HP*β*CyD in two distinct molar ratios (1:1 and 2:1, HP*β*CyD/IBU), and the structure and properties of these HP*β*CyD/IBU inclusion complex NFs were examined by utilizing appropriate characterization methods. It is also worth noting that electrospinning was performed using aqueous solutions of the HPβCyD/IBU inclusion complex, which has a significant benefit because HP*β*CyD makes IBU water soluble. As a result, only water can be used to electrospin HP*β*CyD/IBU inclusion complex NFs, whereas hazardous organic solvents and hydrophobic drugs are employed to dissolve the polymeric matrix and electrospin polymer/drug-based fast-dissolving NFs. When exposed to water or wetted with fake saliva, the HP*β*CyD/IBU NFs demonstrated a very quick dissolving nature, indicating that such electrospun HP*β*CyD/IBU NFs have potential as a fast-dissolving oral drug delivery method.

In another study, Celebioglu et al. [109] developed electrospun hydrocortisone/cyclodextrin complex NFs for a fast-dissolving oral drug delivery method. They used hydrocortisone as a model active pharmaceutical ingredient for electrospinning polymer-free NFs of a hydrocortisone/CyD inclusion complex to develop an orally fast-dissolving DDS. Hydrocortisone, a corticosteroid, is widely used in medicine because of its anti-inflammatory and immunosuppressive properties. Although hydrocortisone is a water-insoluble hydrophobic drug with limited bioavailability, its water solubility can be considerably improved by the inclusion of CyD complexes. For example, CyDs are being investigated for oral hydrocortisone disease therapy, formulation of aqueous ophthalmic hydrocortisone solutions, and oral pediatric hydrocortisone solutions. Electrospinning of hydrocortisone-containing polymeric NFs has also been described for cutaneous wound healing and regulated drug delivery. For oral drug administration, Topuz et al. [113] explored fast-dissolving antibacterial NFs of cyclodextrin/antibiotic inclusion complexes. The authors devised an oral antibiotic delivery method based on electrospun CyD@antibiotics NFs. They prepared aqueous HP*β*CyD solutions with various antibiotics and adjusted the CyD concentration to generate ultrafine fibers for each antibiotic system. The investigation revealed the antibacterial properties of the fibers employing Gram-negative bacteria in a disk diffusion experiment (*E. coli*).

### 5.2. Implantation

The ability to implant directly at the site of action is a significant advantage of nanofibrous delivery methods, as it reduces the systemic toxicity of the implanted drug [114]. Several stimuli-responsive nanofibrous devices have been developed to improve the specificity of medication action [110]. The ease with which polymeric fibers may be manipulated as macroscopic bulk materials suggests that they can be used as implanted local drug delivery platforms [103]. Wsoo et al. [114] used electrospun cellulose acetate (CA)/polycaprolactone NFs to produce a prolonged drug delivery method. The major goal of their research was to develop a novel implantable DDS (IDDS) based on electrospun polymer NFs. The implants in this method could be utilized to provide vitamin D_3_ over a long period through subcutaneous tissues. The IDDS was created using electrospun CA and PCL NFs. The implant core was composed of a drug-loaded CA NF (CA + Vit.D_3_) wrapped in a PCL membrane rate-limiting layer (CA + Vit.D_3_/PCL). In vitro cytotoxicity tests revealed that HDFa cells had high cell survival and proliferation in the model including CA NFs encapsulated in sintered PCL NFs. Based on the results and ease of use of the technologies described in their study, the created implant may be appropriate for long-term medication administration by being implanted in subcutaneous tissues.

Zong et al. [110] investigated the use of cisplatin (CDDP)-loaded mucoadhesive NFs for local cervical cancer treatment in mice. In their study, mucoadhesion, in vitro and in vivo release profiles, and biodistribution of CDDP-loaded PEO/PLA composite electrospun NFs were evaluated. An orthotopic cervical cancer mouse model was developed and utilized to evaluate the effectiveness and safety of the drug-loaded NFs. The goal of their study was to investigate the potential and practicality of a vaginal DDS based on NFs for local chemotherapy against cervical cancer. By directly injecting U14 cells into the vaginal submucosa near the cervix in situ, an orthotopic mouse model of cervical/vaginal malignancies was effectively generated. In vivo trials revealed that the CDDP/fiber remained in the vaginal tract throughout the test period without leakage. Moreover, vaginal implantation of the CDDP/fiber resulted in preferred partitioning of CDDP in the vaginal tract, reasonable distribution in the rectum, uterus, and tumor, and a very low concentration in peripheral organs, yielding a good balance between safety and antitumor efficacy compared to other methods. To summarize, drug-loaded NF mats are a promising dosage form for the local treatment of cervical/vaginal malignancies. It is appropriate for the treatment of unresectable cervical/vaginal malignancies or as an adjunct to surgical resection of these tumors.

Irani et al. [115] devised a new biocompatible DDS based on CS/temozolomide (CS/TMZ) NP-loaded PCL-PU NFs for long-term TMZ administration. In their study, CS/TMZ NPs were effectively produced via ionic gelation, and the NPs were embedded in PCL-Diol-b-PU NFs. Gold NPs were effectively deposited on the surface of the NFs. The homogeneous CS/TMZ NPs had an average particle size of 70 nm. The NFs produced had a high TMZ encapsulation efficiency. The zero-order kinetic model was used to obtain continuous TMZ release for 30 days from both the CS/TMZ-loaded PCL-Diol-b-PU and gold-coated NFs. Gold treatment on the surface of CS/TMZ-loaded PCL-Diol-b-PU/TMZ NFs increased their cytotoxicity. These findings indicate that gold-coated CS/TMZ-loaded PCL-Diol-b-PU NFs can be utilized as appropriate drug delivery implants to administer TMZ for the treatment of glioblastoma cancer.

Samadzadeh et al. [103] described implanted smart hyperthermia NFs with switchable, regulated, and sustained drug delivery. In their work, a smart hyperthermia NF with simultaneous heat production and dual-stage drug release capability in response to AMF on–off switching was created for enhanced hyperthermic chemotherapy. It was discovered that MET-MET@MSNs-MNFs showed a combination of early fast and late extended drug discharge. The metabolic activity of B16F10 cutaneous melanoma cells cultured with all types of MNFs was reduced by exposure to a magnetic field for 300 s on the second and third days. Because of the combined effects of heat and dual-stage drug release, MET-MET@MSNs-MNFs showed higher cytotoxicity than MET-MNFs and MET@MSNs-MNFs (*p* < 0.05). These findings show that the proposed two-stage drug discharge method combined with heat is preferable to conventional chemotherapy regimens and may effectively cause cytotoxicity via a synergistic effect over a reasonably long period.

Li et al. [116] demonstrated extremely bioadhesive-implanted NFs that continually release cytostatic and anti-inflammatory drugs to prevent peritoneal adhesions (Figure 19). For peritoneal adhesion therapy, the NFs were created using a core–sheath NF filled with hydrophobic HCPT in the sheath and hydrophilic diclofenac sodium (DS) in the core. Electrospinning of mPEG-b-PLGA and dextran emulsion with HCPT in the oil phase and DS in the aqueous phase produced co-loaded NFs. First, an ultraviolet-ozone (UVO) treatment was employed to boost the membrane’s bioadhesion, which improved the membrane’s physical isolation effect. Second, owing to the well-designed core–sheath structure, the release behaviors of both HCPT and DS were continuous and sustained for several days. Significantly, the UVO-treated and dual-drug-coloaded membranes exhibited the highest anti-adhesion efficiency.

### 5.3. Skin Treatment

The distribution of medicines using NFs is based on the simple principle of a higher drug dissolution rate owing to the increased surface area of the drug and the carrier. Because the drug molecules are entrapped inside the polymer framework, NFs function as a controlled drug delivery method. These drug-infused NFs can be placed on the skin to aid in wound healing or for easy drug release for systemic or local therapeutic activity. While the cosmetic use of pharmacological substances on the skin is a key problem, NF-based formulations have shown some potential applications [117]. Rezaei et al. [111] published the results of a drug release study using vitamin C (VC)-loaded SA/PEO NFs for the treatment of a skin disease. VC was integrated into PEO/SA NFs using two distinct electrospinning settings (core shell and blended) to create a medication delivery system for pigmented purpuric dermatosis (PPD) therapy. The results showed that the quantity of VC and SA in the electrospinning solution affected the viscosity and electrical conductivity of the solution, as well as the final fiber diameter. These demonstrated the effective integration of VC into the NFs. The degradation rate was enhanced by adding SA and VC to the PEO NFs. According to the drug release research results, the core-shell NFs had a lower release rate than the blended NFs because of the presence of VC further from the surface of the NFs. The investigation of the skin absorption of NFs also revealed that core-shell NFs have slower VC penetration than blended NFs. The drug release rate from the stretched core-shell NFs was also somewhat greater during the initial release, according to the findings. Overall, the findings showed that the core-shell NF with a more controlled release behavior of VC has the potential to be used as a drug delivery vehicle in the treatment of PPD.

Yang et al. [118] created multifunctional CS/PCL NFs with variable dual-drug release for wound healing (Figure 20). Electrospinning-based wound dressings with multifunctional features such as hemostasis promotion, antimicrobial, medication release, and therapeutic effects are gaining popularity in military and civilian trauma treatment. The authors created lidocaine hydrochloride (LID) and mupirocin-loaded CS/PCL (CSLD-PCLM) scaffolds with various wound dressing functionalities. The scaffolds obtained an NF structure using the dual spinneret electrospinning process, which improved the interfacial contact between the scaffold and blood cells and demonstrated good blood coagulation capacity. The scaffolds loaded with LID and mupirocin showed fast LID release and prolonged mupirocin release. The antibacterial efficacy of the CSLD-PCLM scaffold containing mupirocin was exceptional. Furthermore, in a full-thickness skin defect model, the scaffold remarkably improved wound healing by allowing complete re-epithelialization and collagen deposition. The dual spinneret electrospinning technique was used to successfully develop CSLD-PCLM NF dressings with multifunctional properties such as excellent hydrophilicity, absorbing capacity, mechanical properties, cytocompatibility, and antibacterial properties. This scaffold also showed excellent in vitro blood coagulation capacity. Furthermore, LID and mupirocin were encapsulated in composite NFs of varying compositions to enable fast LID release and prolonged mupirocin release.

## 6. Conclusions

In this review, we discuss on-demand DDSs using NFs, which are making headlines in the pharmaceutical industry. Because of their distinctive porous structures and large surface-to-volume ratio, they are suitable for a wide range of applications. Nanostructured drug delivery architectures are potential options for enabling efficient and novel drug delivery. Electrospinning is the most versatile method for creating NFs with a wide range of characteristics. Efforts to construct electrospun polymer NF scaffolds for neurons, tissues, the skin, and bone are among the potential medicinal uses. The selection of the polymer(s) (e.g., water-soluble polymer for immediate drug release, swellable or degradable polymer for prolonged drug release, and stimuli-responsive polymer for stimulus-activated drug release), solvent, and electrospinning setup form the foundation for the development of NFs with the desired drug release properties (e.g., single nozzle and co-axial electrospinning) (Table 1). The features of NF-based DDSs can also be modified by making precise changes to the process and ambient parameters. The composition and structure of NFs have a major impact on drug release. Furthermore, the route of drug administration and properties of the specific pharmacological material are critical variables, as they can influence drug release. Therefore, NFs with the necessary drug-release profile for every specific DDS are frequently made to order.

The development of NFs should prioritize the generalization of the technique, large-scale production, clinical trials, and innovative strategies for the creation of medicated NFs. Aside from conventional electrospinning, there are numerous techniques for fabricating NFs with unique designs, such as blow-jet spinning, centrifugal spinning, microfluidic spinning, bubble electrospinning, and a combination of several current techniques. In terms of regulated structure and architecture with high complexity, these techniques may offer more efficient and effective pathways to high quality NF structures in a timely manner. In this review, we focused on the electrospinning method for fabricating NFs because the diameter of NFs can be easily controlled and obtained in nanometers when compared to the other techniques. New technologies, such as electrospraying, will most likely focus on the fabrication of core-shell structures from a mix of spinnable and non-spinnable solutions. In electrospraying, the jet can be kept in a continuous form to break up into droplets rather than produce fibers. The other technique is blown solution spinning, which allows for the fabrication of fibers in the nanometer range without the use of a high voltage gradient, which is useful when working with cells or other bio-systems. Furthermore, this fabrication process is carried out under atmospheric pressure, does not require harsh chemical conditions, and can be carried out at room temperature. Nonetheless, blow spinning is still in its early stages, and creating customized nozzles is difficult. Melt blown spinning is a method of preparing fibers without the use of a solvent in which the melt is extruded from a hole in the mold by air flow. Melt blown spinning has several flaws, including degradation during fabrication due to heating. The polymer can degrade if the temperature is not properly controlled.

Several limitations must be overcome before NFs can be considered for clinical trials. The most important is the requirement for subsequent removal surgery because modern local DDSs are not biodegradable. Thus, the possible threat of the remaining solvents is the next problem. Typical solvents employed in the electrospinning precursor process, such as dimethylformamide, chloroform, dichloromethane, tetrahydrofuran, 1,1,1,3,3,3 hexafluoro-2-propanol, and TFE, are hazardous to polymeric NFs. Additional issues, such as industrial scale-up of LDDSs, drug-matrix incompatibility, and optimal drug release, need to be addressed before this technology is implemented in clinical practice. Although NFs are potential drug delivery technologies for obtaining specific drug release patterns, further studies are required to demonstrate their commercial and clinical applications.

## Figures and Tables

**Figure 1 nanomaterials-11-03411-f001:**
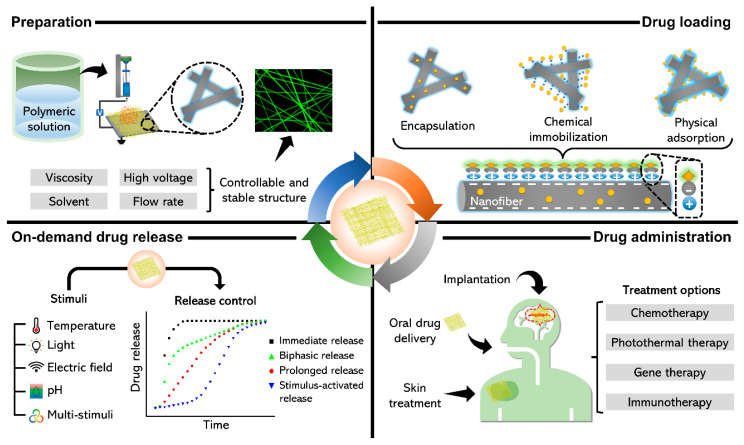
Schematic of on-demand drug delivery systems using stimuli-responsive NFs.

**Figure 2 nanomaterials-11-03411-f002:**
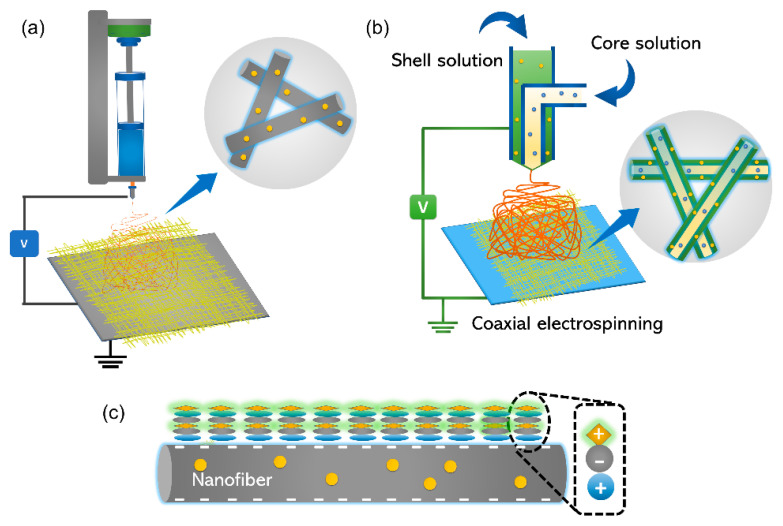
Schematic of preparation of (**a**) blended NFs using electrospinning, (**b**) core-shell NFs using coaxial electrospinning, and (**c**) layer-by-layer assembly as DDSs.

**Figure 3 nanomaterials-11-03411-f003:**
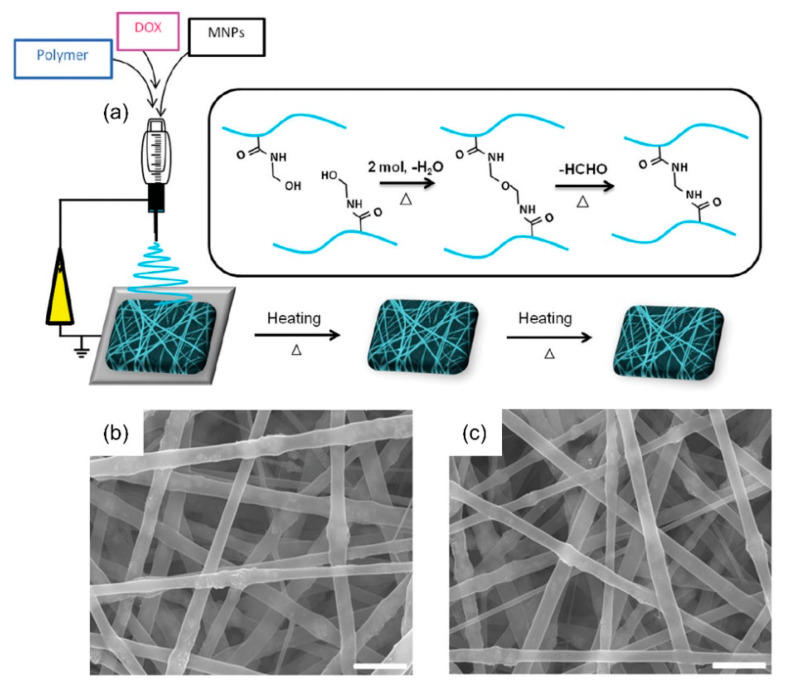
(**a**) Preparation of thermally crosslinkable temperature-responsive NFs by blended electrospinning technique. Scanning electron microscopy (SEM) images of DOX/MNP NFs (31 wt% of MNPs and 0.18 wt% of DOX) (**b**) before and (**c**) after crosslinking (scale bars are 1 μm). Reproduced with permission from [51], Copyright © 2021, John Wiley & Sons, Inc.

**Figure 4 nanomaterials-11-03411-f004:**
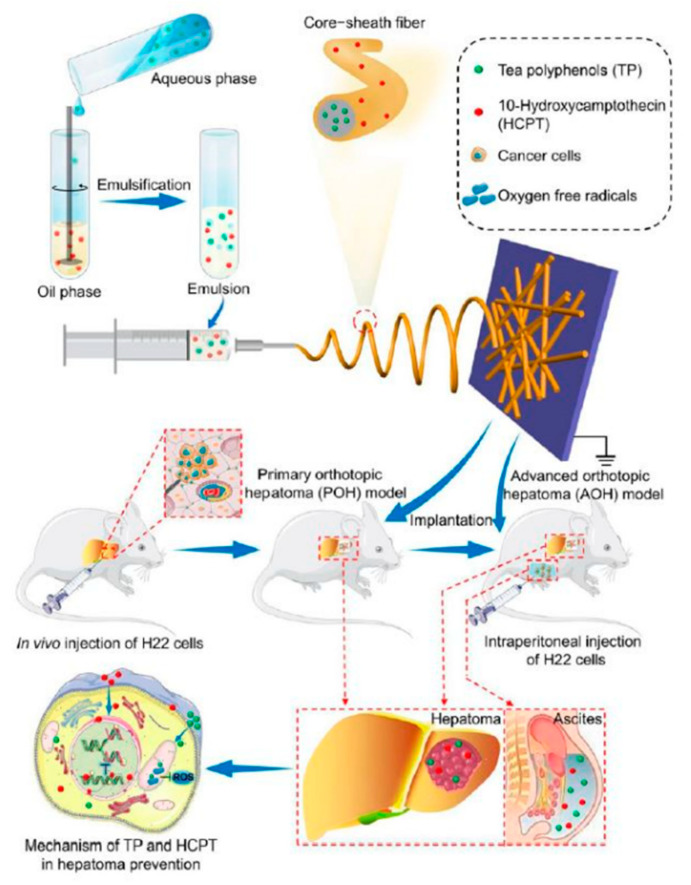
Core–shell NFs for sequential drug delivery of HCPT and TP. Reproduced with permission from [57], Copyright © 2021, American Chemical Society.

**Figure 5 nanomaterials-11-03411-f005:**
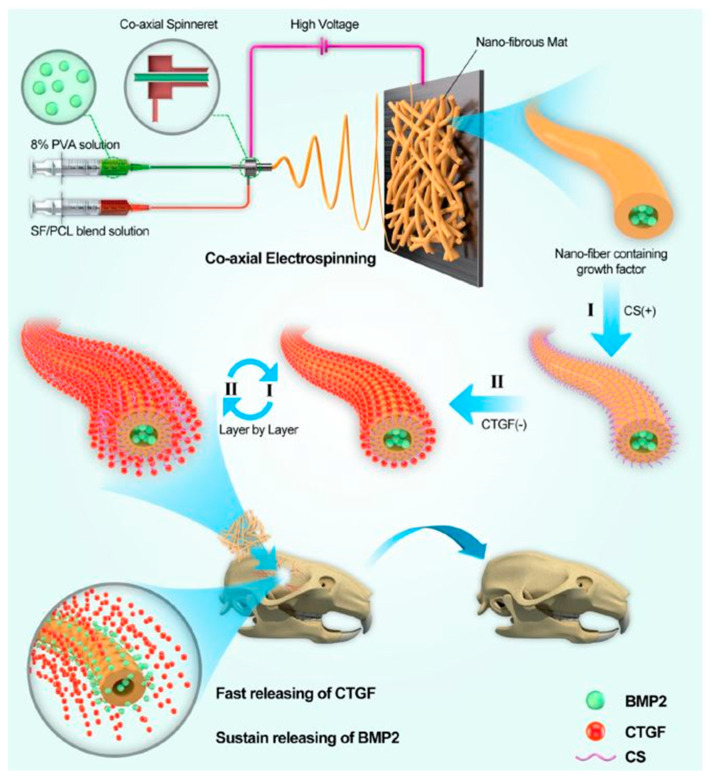
Layer-by-layer assembly of core-shell NFs for controlled co-delivery of growth factors for bone tissue engineering. Reproduced with permission from [66], Copyright © 2021, American Chemical Society.

**Figure 6 nanomaterials-11-03411-f006:**
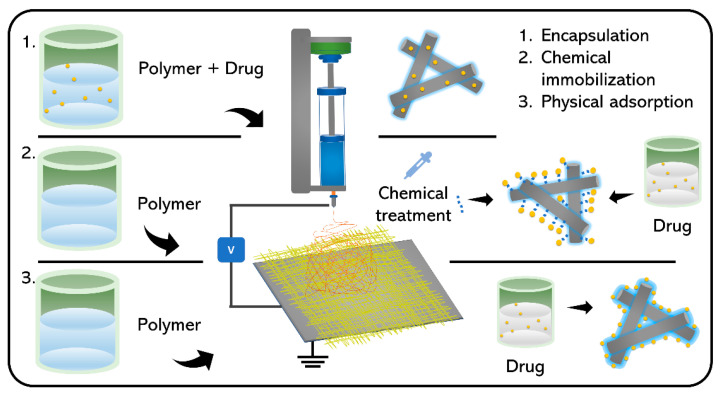
Schematic of various methods of drug loading.

**Figure 7 nanomaterials-11-03411-f007:**
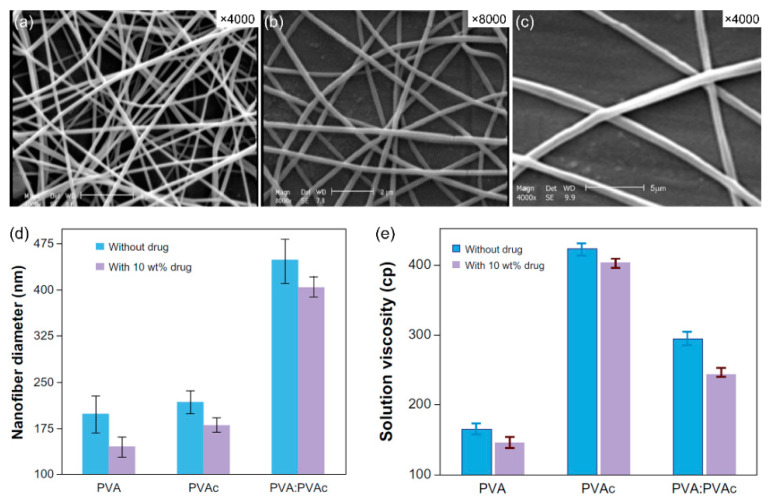
SEM photographs of CipHCl loaded electrospun nanofibers (**a**) PVA, (**b**) PVAc, and (**c**) 50:50 blend of PVA/PVAc. Effect of CipHCl on (**d**) the diameter of nanofibers and (**e**) the solution viscosity. Reproduced with permission from [68], Copyright © 2021 Jannesari et al., publisher and licensee Dove Medical Press Ltd., Macclesfield, United Kingdom.

**Figure 8 nanomaterials-11-03411-f008:**
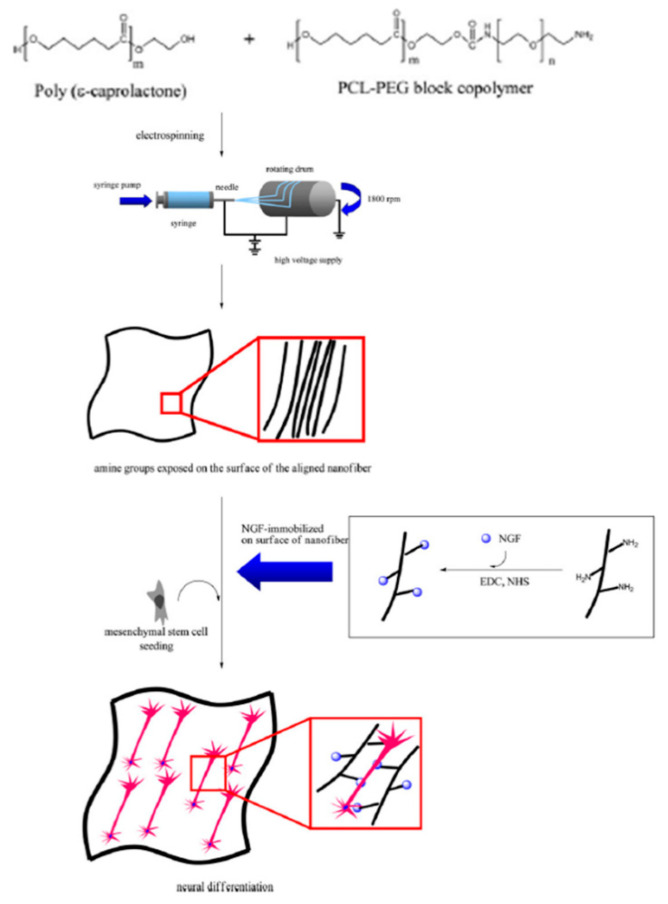
Schematic overview of preparing NGF–conjugated aligned nanofibers to differentiate rMSCs into neuron cells. Reproduced with permission from [71], Copyright © 2021 Acta Materialia Inc. Published by Elsevier Ltd., Amsterdam, The Netherlands.

**Figure 9 nanomaterials-11-03411-f009:**
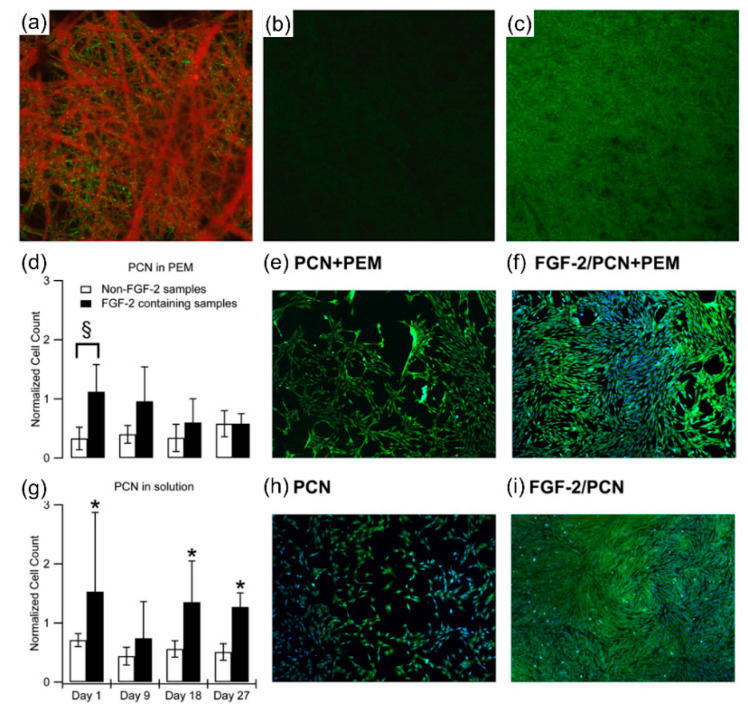
(**a**) Confocal microscopy images using ×63 objective of rhodamine-modified chitosan fiber network (red) with FGF-2^LB^/PCN complexes adsorbed (green). (**b**,**c**) Confocal microscopy images (using ×20 objective) of FGF-2^LB^/PCN fibers and FGF-2^LB^/PCN + PEM fibers after 30 days of release at 37 °C. (**e**,**f**,**h**,**i**) Representative fluorescence microscopy images of MSCs stained with DAPI (nuclei) and calcein-AM (cytoplasm). (**d**–**f**) Cells cultured on TCPS coated with chitosan, PCNs (or FGF-2/PCN complexes), one bilayer of TMC–heparin PEM, and fibronectin. (**g**–**i**) Cells cultured on fibronectin-coated TCPS with PCNs (or FGF-2/PCN complexes) delivered in solution. § in (**d**) indicates statistically different results for the two conditions. * in (**g**) indicates that the FGF-2/PCN-complexes preconditioned for 1, 18, and 27 days in solution result in cell densities that are statistically different from all other conditions in both (**d**) and (**g**) that correspond to the same preconditioning time (n = 3, *p* < 0.05). Reproduced with permission from [78], Copyright © 2021 Acta Materialia Inc. Published by Elsevier Ltd., Amsterdam, The Netherlands.

**Figure 10 nanomaterials-11-03411-f010:**
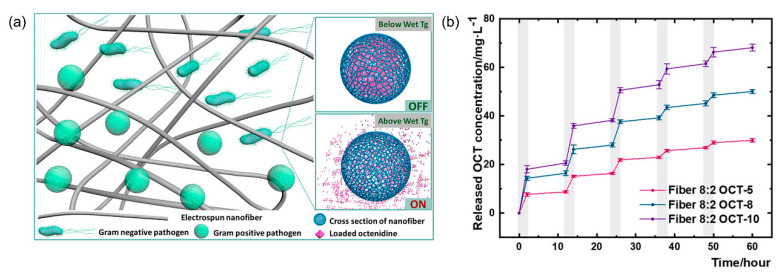
(**a**) Schematic hypothesis of T*_g_*−triggered OCT release from NFs. (**b**) In vitro pulsewise drug release from NFs. Reproduced with permission from [90], CC BY−NC−ND license, Copyright © 2021 The Authors, Published by American Chemical Society.

**Figure 11 nanomaterials-11-03411-f011:**
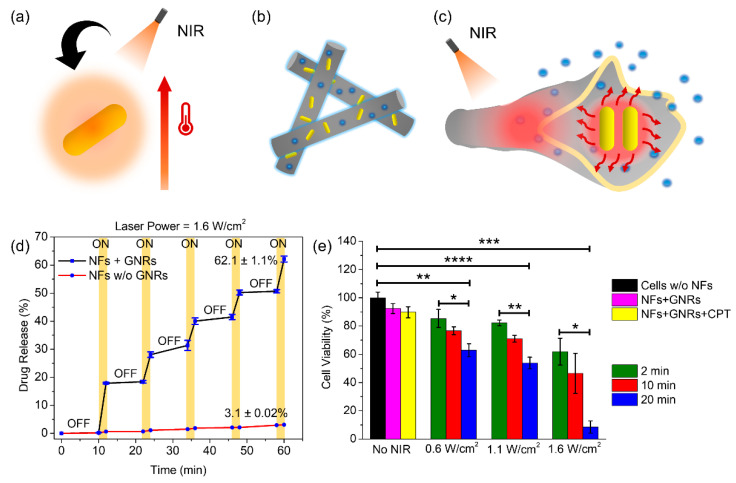
Schematic of (**a**) heat generated by GNRs upon NIR irradiation, (**b**) NFs encapsulating drug and GNRs after electrospinning, and (**c**) drug release due to shrinkage of NFs upon NIR irradiation. (**d**) Pulsatile drug release from NFs through the cyclic on–off of NIR light irradiation at different time intervals. (**e**) Cell viability of U87 cells due to CPT release from the NFs upon NIR irradiation at different time intervals. Reproduced with permission from [80], CC BY license, Copyright © 2021 The Authors, Published by MDPI. * *p* ≤ 0.05, ** *p* ≤ 0.01, *** *p* ≤ 0.001, **** *p* ≤ 0.0001.

**Figure 12 nanomaterials-11-03411-f012:**
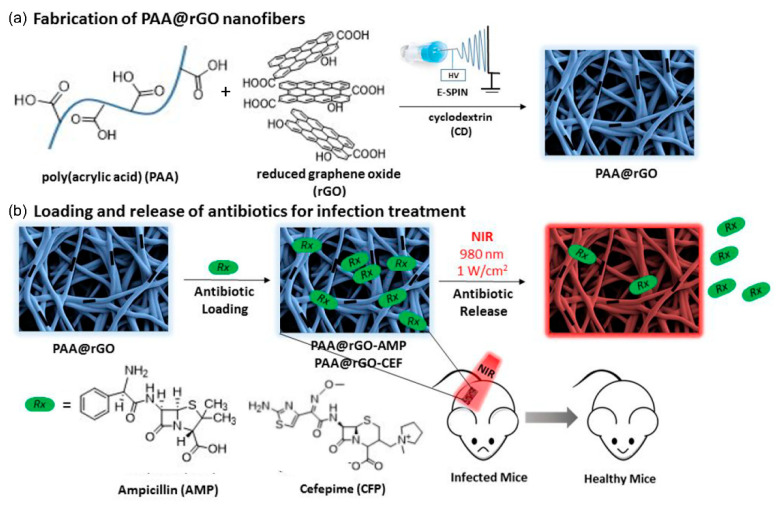
Schematic of (**a**) electrospinning of rGO-loaded PAA NF mats followed by (**b**) loading with different antibiotics and photothermal triggered antibiotic release. Reproduced with permission from [91], Copyright © 2021, American Chemical Society.

**Figure 13 nanomaterials-11-03411-f013:**
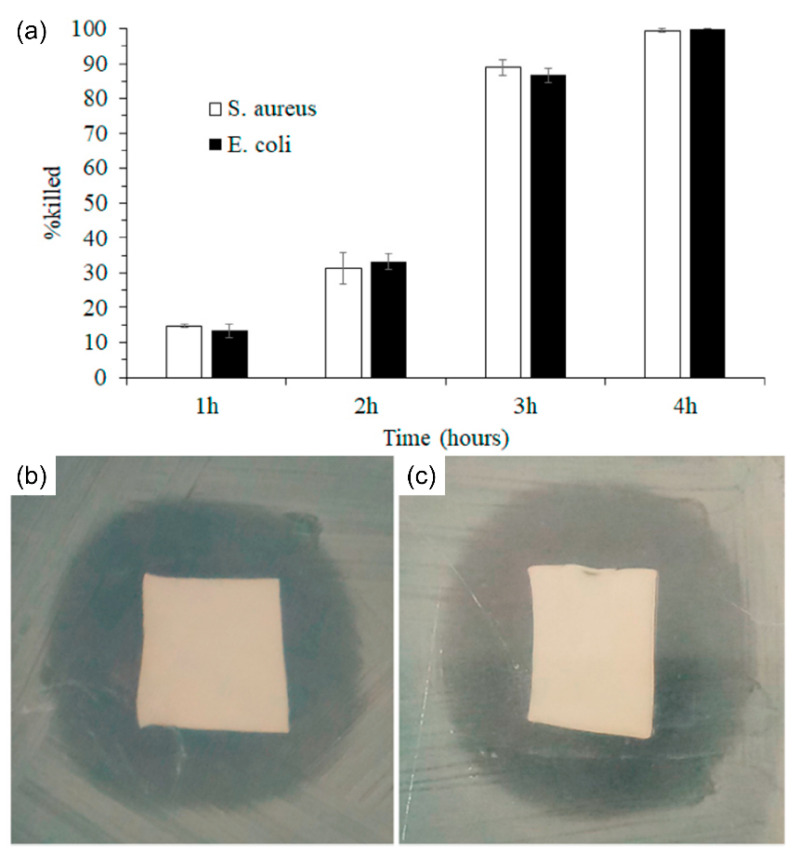
(**a**) Antibacterial activity of PVA/PAA10*-C samples against *S. aureus* and *E. coli* and inhibition zone of PVA/PAA10*-C samples against (**b**) *E. coli* and (**c**) *S. aureus*. Reproduced with permission from [96], Copyright © 2021, Elsevier.

**Figure 14 nanomaterials-11-03411-f014:**
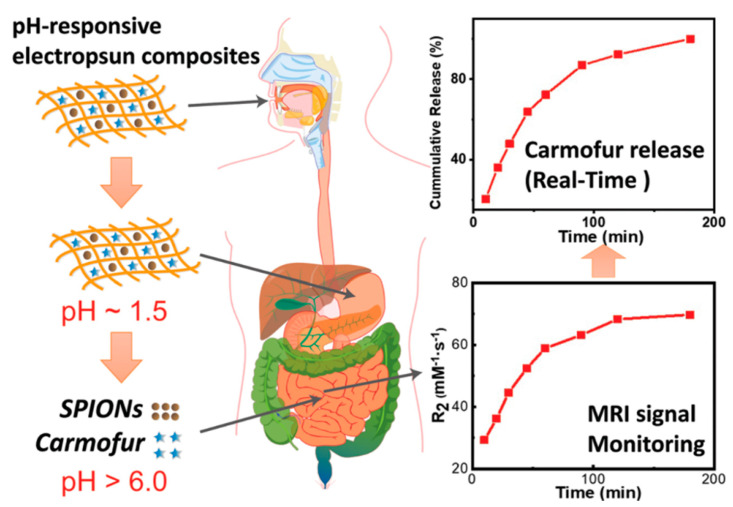
Strategic underpinning nanoplatform design of pH-responsive NFs allowing MRI monitoring of drug release. Reproduced with permission from [101], CC BY license, Copyright © 2021, The Royal Society of Chemistry.

**Figure 15 nanomaterials-11-03411-f015:**
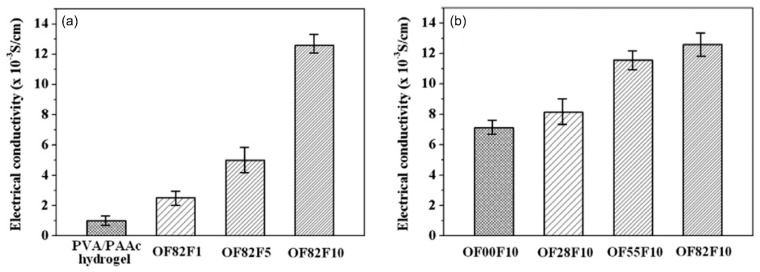
Variations in electrical conductivity of nanofibers depending on (**a**) content of oxyfluorinated MWCNTs and (**b**) oxyfluorination condition for MWCNTs. Reproduced with permission from [102], Copyright © 2021, Elsevier.

**Figure 16 nanomaterials-11-03411-f016:**
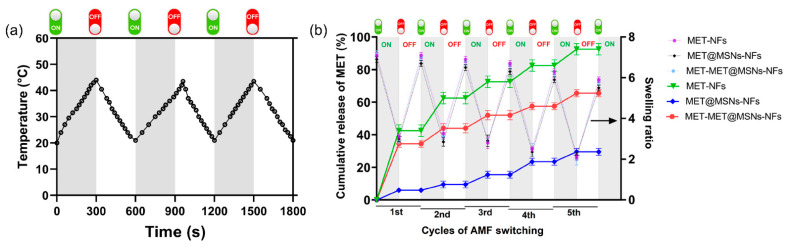
(**a**) Heating and cooling profile of the MET-MET@MSNs-MNFs in response to alternating switching of AMF (**b**) ‘ON-OFF’ switchable and reversible heat profile and swelling ratio of the MNFs with increasing ‘ON-OFF’ switching cycle of AMF, and MET release pattern corresponding to reversible swell-shrink property in response to temperature changes. Reproduced with permission from [103], Copyright © 2021, Elsevier.

**Figure 17 nanomaterials-11-03411-f017:**
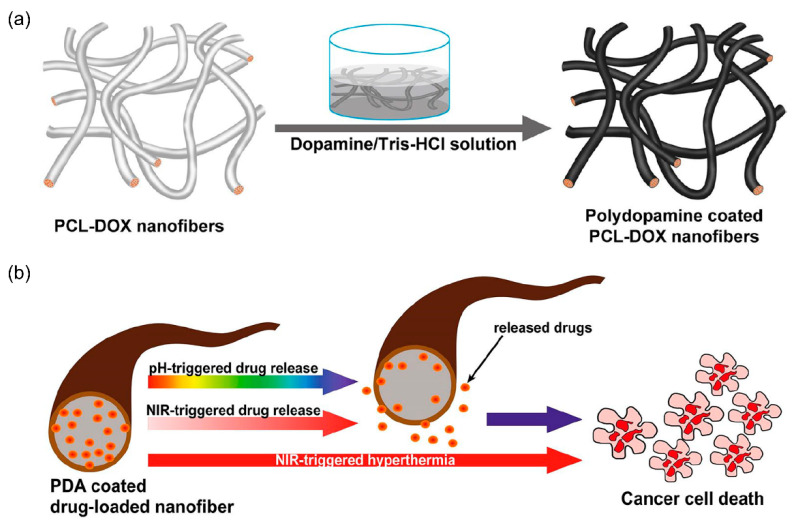
(**a**) Schematic of the fabrication procedure of the PDA modified PCL-DOX nanofibrous mat. (**b**) Schematic of cell death mechanism. Reproduced with permission from [94], CC BY license, Copyright © 2021 The Authors, Published by Springer Nature.

**Figure 18 nanomaterials-11-03411-f018:**
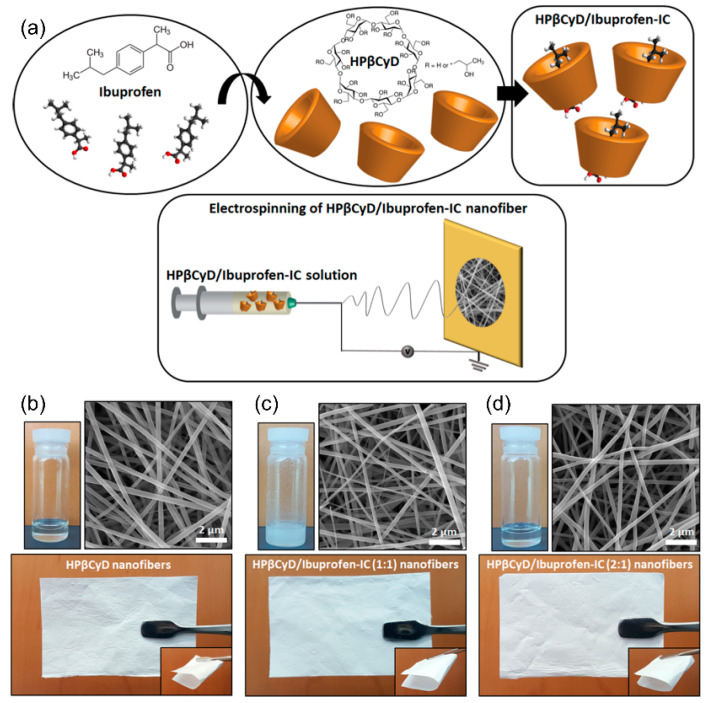
(**a**) Schematic of the inclusion complex formation between ibuprofen and HP*β*CyD molecules and electrospinning of HP*β*CyD/ibuprofen-IC NFs. Photographs of electrospinning solutions and the resulting electrospun nanofibrous webs and representative SEM images: (**b**) pure HP*β*CyD NFs, (**c**) HP*β*CyD/ibuprofen-IC NFs (1:1), and (**d**) HP*β*CyD/ibuprofen-IC (2:1) NFs. Reproduced with permission from [112], Copyright © 2021, American Chemical Society.

**Figure 19 nanomaterials-11-03411-f019:**
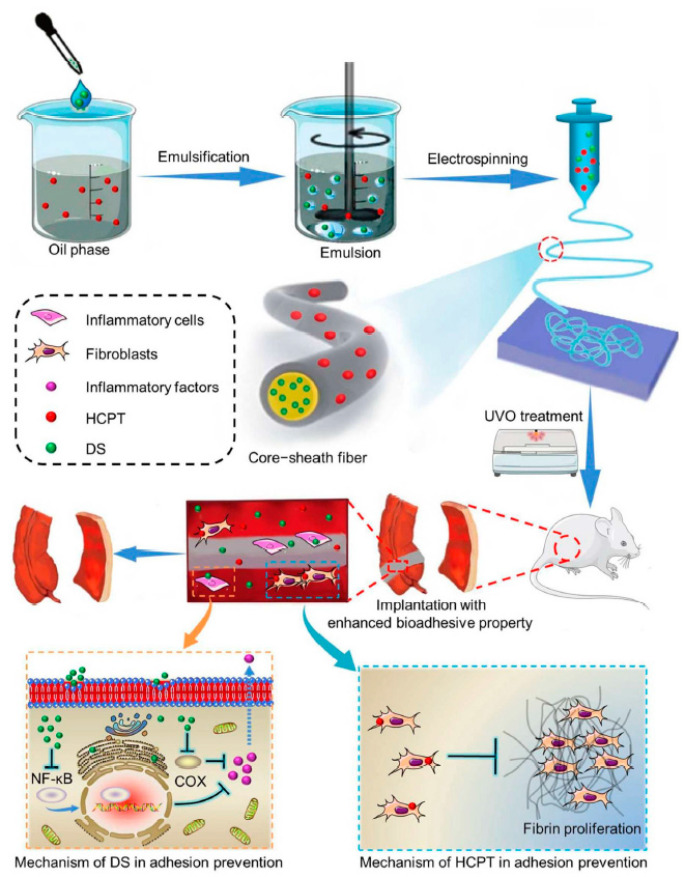
Schematic of preparation and anti-adhesion of HCPT and DS co-loaded NFs for implantation. Reproduced with permission from [116], Copyright © 2021, American Chemical Society.

**Figure 20 nanomaterials-11-03411-f020:**
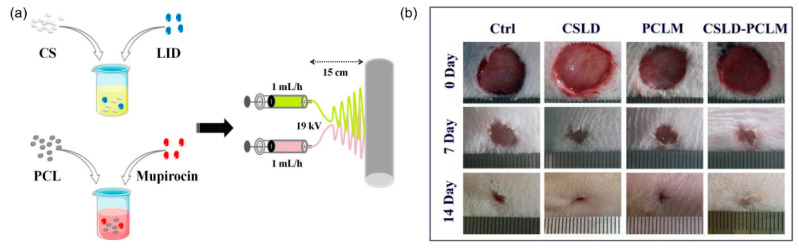
(**a**) Fabrication process of CSLD-PCLM NF scaffolds. (**b**) Wound healing effect of CSLD-PCLM NF scaffolds in a full-thickness skin defect model. Reproduced with permission from [118], Copyright © 2021, American Chemical Society.

**Table 1 nanomaterials-11-03411-t001:** Types, drug loading mechanism, drug administration, stimuli and release types, and composition of NFs.

No.	Type	Loading Mechanism	Stimuli	Drug	Administration	Release Type	Composition	Ref. No.
1	Blended	Encapsulation	Light	RhB	Implantation	Rapid and controlled release	PLLA	[50]
2	Blended	Encapsulation	Magnetic field	DOX		Switchable release	Poly (NIPAAmco-HMAAm)	[51]
3	Blended	Physical adsorption	Mechanical	Crystal violet	Implantation	Controlled release	P(VDF-TrFE)	[52]
4	Blended	Encapsulation		ART	Transdermal	Sustained release	PCL/Col	[53]
5	Core–shell	Encapsulation		DSS, GENS	Wound dressing	Dual-drug release	PVA, PAN	[56]
6	Core–shell	Encapsulation		TP, HCPT	Implantation	Sequential release	Dextran, (mPEG-b-PLGA)	[57]
7	Core–shell	Encapsulation		DOX	Implantation	Controlled and sustained release	PLCL/gelatin	[58]
8	Core–shell	Encapsulation		5-Fu	Implantation	Prolonged release	PCL, PVP	[59]
9	LbL	Encapsulation		TTC	Wound dressing	Controlled release	PCL	[64]
10	LbL	Physical adsorption	pH	IBU		Controlled release	PLGA	[65]
11	LbL	Physical adsorption		CTGF, BMP2	Implantation	Sustained and fast release	PVA, SF/PCL	[66]
12	Blended	Encapsulation		CipHCl	Wound dressing	Controlled release	PVA/PVAc	[68]
13	Blended	Encapsulation		GAPDH siRNA	Implantation	Controlled release	PCL	[69]
14	Blended	Encapsulation		FBF		Controlled release	PLGA/gelatin	[70]
15	Blended	Chemical immobilization		NGF	Wound dressing	Controlled release	PCL/PEG	[71]
16	Blended	Chemical immobilization		Procion Blue		Controlled release	PVA	[72]
17	Blended	Physical adsorption		DOXY-h	Wound dressing	In vitro drug release	PAA	[76]
18	Blended	Physical adsorption		Daunorubicin	Implantation	Controlled release	PLA	[77]
19	Blended	Physical adsorption		FGF-2	Wound dressing	Controlled release	Chitosan	[78]
20	Blended	Physical adsorption		PTX	Wound dressing	Controlled release	CS/PEO	[79]
21	Blended	Encapsulation	Temperature	IBU		Controlled and switchable release	PNIPAM/PCL	[88]
22	Blended	Encapsulation	Temperature	FITC-dextran		Switchable release	poly(NIPAAm-*co*-HMAAm)	[89]
23	Blended	Encapsulation	Temperature	RhB	Implantation	Controlled release	ERS/PMMA	[82]
24	Blended	Encapsulation	Temperature	OCT	Wound dressing	Controlled release	ERS/PMMA	[90]
25	Blended	Encapsulation	Light	CPT	Implantation	On-demand release	PNIPAM	[80]
26	Blended	Surface functionalization	Light	Ampicillin and cefepime	Wound dressing	On-demand release	PAA/rGO	[91]
27	Core–shell	Encapsulation	Light	MB	Skin treatment	On-demand release	PVP, PVA	[92]
28	Blended	Encapsulation	Light	Levofloxacin	Wound dressing	Controlled release	PVA	[93]
29	Blended	Encapsulation	Light	DOX	Implantation	Controlled release	PCL	[94]
30	Blended	Encapsulation	Light	BTZ	Implantation	Controlled release	PDO	[95]
31	Blended	Encapsulation	pH	Ciprofloxacin	Wound dressing	On-demand release	PVA/PAA	[96]
32	Blended	Encapsulation	pH	Ciprofloxacin		Controlled release	poly(VBA-co-VBTAC)	[99]
33	Cellulose	Encapsulation	pH	DOX		Sustained release	PEI	[100]
34	Forcespinning	Encapsulation	pH	DOX	Implantation	Controlled release	PCL	[97]
35	Blended	Physical adsorption	pH	DOX, R6G		Controlled release	PCL	[98]
36	Blended	Encapsulation	pH	Carmofur		Controlled release	Eudragit	[101]
37	Blended	Encapsulation	Magnetic field	MET	Implantation	Sustained release	poly(NIPAAm-co-HMAAm)	[103]
38	Cellulose	Encapsulation	pH and electric field	IBU		Controlled release	nf-BC/SA	[108]
39	Blended	Encapsulation	pH and temperature	BSA		Controlled release	CTS-g-PNIPAAm/PEO	[107]
40	Blended	Encapsulation		IBU	Oral	Controlled release	HP*β*CyD	[112]
41	Blended	Encapsulation		Hydrocortisone	Oral	Fast release	Hp*β*CyD	[109]
42	Blended	Encapsulation		Antibiotics	Oral		Hp*β*CyD	[113]
43	Core–shell/Blended	Encapsulation		Vit. D_3_	Implantation	Prolonged release	CA, PCL	[114]
44	Blended	Encapsulation		CDDP	Implantation		PEO/PLA	[110]
45	Blended	Encapsulation		TMZ		Controlled release	PCL-Diol-b-PU	[115]
46	Core–shell	Encapsulation		DS, HCPT	Implantation	Sustained release	Dextran, mPEG-b-PLGA	[116]
47	Core–shell/Blended	Encapsulation		VC	Wound dressing	Controlled release	SA/PEO	[111]
48	Blended	Encapsulation		LID, mupirocin	Wound dressing	Dual-drug release	Chitosan/PCL	[118]

## Data Availability

Not applicable.

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
