# Peer review of "On-Demand Drug Delivery Systems Using Nanofibers"

_nanomaterials, 2021, doi:10.3390/nano11123411_

Round 1

Reviewer 1 Report

This is an interesting and well-written manuscript that should be pulished in the Journal after the following minor changes:

1) I encourage the Author to add more Figures, such as e.g. in the case of duscussing encapsulation part (subsection 3.1)

2) The Authors should refere to the previous reviews on the similar topics.

3) Impact of this work should be further highlighted.

4) It would be nice to present some graphical representation of the contents of the introduction of part 2 "2. Types and preparation of NFs as DDSs" in page 3.

5) Some drawings (including chemical structures) on figures need to be enlarged, such as in the case of e.g. Figure 8.

Author Response

We are thankful to the editor and reviewers for peer review and valuable comments for our manuscript. Based on the comments of the reviewers, we have revised our manuscript and modified as following.

Please read the attached file.

Reviewer 2 Report

This paper includes an intensive review about drug delivery application based on nanofibers trend. This review paper is worth to be published after addressing some minor comments, as follows:

1- The authors should make one section "or paragraph" about other techniques to generate nanofibers rather than electrospinning, such as solution blowing or melt blown spinning, due to the higher scale of fabricated mats within other techniques.

2- The electrospinning is extensively used in drug delivery applications, but the scaling up is an issue. So, it is better to add one paragraph about scaling up trials of electrospinning such as needless electrospinning "Nanospider".

3- One SEM or HRTEM with encapsulated drugs "or nanoparticles drugs" is recommended to add it. You can choose anyone with copyrights clearance of zero fees.

4- One photo of inhibition zone comparison for anti-bacterial effect is highly recommended. Also, you can choose anyone with copyrights clearance of zero fees.

5- Figure 10 should have only discrete values without connecting straight lines between mean points. At least, you can make it dotted to show the behavior but not straight lines as it may not be fitted experimentally "you don't know the transient behavior between on-off states".

Author Response

(The authors gave the same response as above.)
